# Assessment of Land Ecological Security from 2000 to 2020 in the Chengdu Plain Region of China

**Lindan Zhang** [1,2,†], **Wenfu Peng** [1,2,*,†] **and Ji Zhang** [1,2]

1   The Institute of Geography and Resources Science, Sichuan Normal University, Chengdu 610068, China; zld_133505@163.com (L.Z.); aizaigongyuanqian@gmail.com (J.Z.)
2   Key Lab of Land Resources Evaluation and Monitoring in Southwest, Ministry of Education, Chengdu 610068, China
*   Correspondence: pwfzh@126.com
†   These authors contributed equally to this work.

**Abstract:** The purpose of land ecological security (LES) assessment is to evaluate the influence of land use and human activities on the land ecosystem. Its ultimate objective is to offer decision-making assistance and direction for safeguarding and rejuvenating the well-being and effectiveness of the land ecosystem. However, it is important to note that there are still significant uncertainties associated with current land ecological safety assessments. This paper presents a comprehensive evaluation model that combines the strengths of subjective and objective weighting methods. The model is built upon an index system developed using the Pressure-State-Response (PSR) framework. To verify the level of LES, theThe results of classifying the total ecosystem service valueTotal Ecosystem Service Value are utilized to verify the level of LES. Furthermore, spatial distribution patterns of regional land ecological safety levels are analyzed using statistical techniques, such as *Moran's I*, Mann–Whitney *U*-test, and Kruskal–Wallis *H*-test. The findings indicate that: (1) theThe evaluation model developed in this paper achieves a validation accuracy of 75.55%, indicating that it provides a more accurate reflection of the level of land ecological safety in the region; (2) The ecological security index is generally safe, with a mean value in the moderate safety range. It experienced a turning point in 2010, showing initial deterioration followed by improvement, mainly due to the transition between unsafe and relatively safe zones. (3) The level of economic development, topography, and urban—rural structure are significant factors influencing the spatial concentration of LES in the region, ultimately shaping the spatial pattern of LES in the Chengdu Plain region.

**Keywords:** combination weights; PSR; ecological safety assessment; spatial pattern analysis



## 1. Introduction

Ecological security refers to the state in which ecosystems can maintain their healthy functioning and sustain their intrinsic value while also meeting the sustainable development needs of human society [1]. Land, as a crucial component of ecosystems, plays a significant role as the material carrier for ecological functions. The health of regional ecosystems is largely dependent on the condition of the land. However, in recent years, rapid industrialization and urbanization have resulted in the conversion of substantial ecological land into artificial surfaces. This conversion has led to changes in the structure and functionality of regional ecological environments, ultimately posing a significant threat to regional ecological security [2,3]. The irrational use of land resources has led to ecological challenges, which in turn have hindered national and regional development [4]. As a result, the issue of land ecological security (LES) has garnered significant attention from scholars worldwide, leading to the establishment of a well-defined research framework [5–7]. Currently, assessing the ecological security status serves as the predominant method to understand the state of LES in a particular region. This assessment typically involves various approaches such as digital model evaluation, ecological model evaluation, landscape

model evaluation, and digital terrain model evaluation [8]. Among these methods, digital model evaluation is widely used and comprises two key components: the construction of an indicator system and the determination of weights [9].

The construction of an indicator system relies on frameworks such as PSR (Pressure-State-Response) and DPSIR (Driving forces-Pressure-State-Impact-Response), which guide the selection of indicators for evaluating a region. The PSR framework focuses on understanding the impact of human activities on ecosystems by examining pressures, states, and responses. It provides an analytical approach to environmental issues, emphasizing how human activities exert pressures on the environment, which in turn affect its state, and prompt societal responses to mitigate pressures and improve environmental conditions. For example, Cheng et al. (2022) developed an indicator system based on the PSR model and utilized the fuzzy comprehensive evaluation method to generate the national ecological security status map for the year 2018 [1]. The reliability of the model was validated using three surrogate indicators. Similarly, Zhao et al. (2019) applied the PSR model to construct an indicator system and investigated the evolving trend of ecological security patterns in Zhangye City [10].On the other hand, the DPSIR framework considers a broader range of factors, including social, economic, and political drivers of environmental change, as well as how these drivers exert pressures, impact environmental states, and trigger societal and policy responses in the context of ecological security. For instance, Shi et al. (2021) established an indicator system based on the DPSIR model and found that social and economic pressures were the primary factors contributing to the conversion of ecological security areas into unsafe areas in Bai Town of Erdaohe [11]. Cui et al. (2021) developed an indicator system using the DPSIR model and emphasized the significance of driving forces and responses in influencing ecological security in the urban agglomeration of the Yangtze River Delta [12]. These models have yielded significant research outcomes due to their multidimensional and comprehensive nature. By following the guidance of the PSR and DPSIR models, researchers can select relevant indicators based on different aspects of the models, enabling a comprehensive and systematic evaluation of the ecological security status in a given region.

Land ecology, as a subfield of ecology, places greater emphasis on the structure, function, and condition of land [13]. In the realm of land ecology, it is important to consider the human-land relationship and the associated ecological values that arise from it [14]. This integration of human-environment dynamics has led to the development of land ecological security assessment [15,16]. Contrasting with land ecology, land ecological security specifically focuses on the capacity of land use patterns to sustainably meet the diverse needs of human society [14]. For instance, Cheng et al. (2022) conducted an assessment of China's ecological land security in 2018 by selecting 18 indicators that encompassed both human demands and land conditions [1]. In the context of rapid urbanization, Zhao et al. (2022) examined the impact of various scenarios of land development on land ecological security, taking into account the changing patterns of human demands [17]. Similarly, in their study on the land ecological security pattern in the Jinan region, Liu et al. (2022) underscored the evaluation of land ecological security status by integrating the ecological values provided by land with human demands [18].

The determination of indicator weights in LES assessment involves both subjective and objective weighting methods. Subjective methods include expert scoring, Analytic Hierarchy Process (AHP), entropy method, and principal component analysis, which are commonly used to evaluate the ecological security status of a region. In recent years, new methods have emerged to address biases resulting from unreasonable weight design. These methods integrate fuzzy mathematics, graph theory, and artificial neural networks into the evaluation process. Examples of such methods include set pair analysis [19], improved projection pursuit model [20], fuzzy reasoning [1], and Bayesian network analysis. These innovative approaches aim to enhance the accuracy and reliability of the assessment process. However, objective methods that solely rely on data variation may not provide an accurate reflection of the influence of data on regional ecological security. Additionally, several

studies have highlighted that even improved objective weighting methods can still produce biased results due to errors in data collection and issues related to sample occurrence probability [2–4,21]. As a result, inconsistencies between evaluation results and the actual situation may arise. Consequently, there is a growing emphasis on weight-determination methods that combine both subjective and objective approaches. These integrated methods aim to produce evaluation results that align more closely with reality [22].

Despite efforts to achieve objectivity in constructing evaluation models, the subjective nature of understanding LES inevitably introduces individual characteristics into the selection of indicator systems by different researchers. Therefore, it becomes necessary to employ methods to verify the reliability of these evaluation models [23]. Model validation allows for the assessment of consistency between subjective concepts of LES and objective measurements of LES, thus establishing the scientific and reliable nature of the LES evaluation model. Cheng et al. (2022) suggested using landscape pattern indices, land use change rates, and primary productivity as objective indicators to independently evaluate the reliability of each dimension (pressure, state, and response) [1]. By employing such objective indicators, the researchers aimed to validate and confirm the robustness of their evaluation model. To verify the ecological security evaluation results, researchers have employed various approaches in different regions. For instance, Zhao et al. (2021) utilized net primary productivity, soil organic carbon, and soil erosion as indicators to validate the ecological security evaluation results of alpine grasslands in the Yellow River Source Region [24]. Hu et al. (2021) used evidence reasoning to establish the basic credibility distribution function of indicators and then assessed the reliability of the ecological security evaluation method in Hunan Province using evidence fusion algorithms and the maximum correlation criterion [25]. Furthermore, Liu et al. (2022) utilized remote sensing ecological indices and high-resolution images to validate the reliability of ecological security evaluation results in the alpine pastoral area of Gannan. Although these studies have examined specific aspects of ecological security from multiple dimensions, they only provide partial validation of the evaluation model, making it challenging to achieve a comprehensive assessment of its overall reliability [26]. Thus, there is a need for a comprehensive dataset that can provide a holistic reflection of the evaluation model's reliability.

The Chengdu Plain region, situated upstream of the Yangtze River Economic Belt, holds great importance in soil and water conservation efforts in China. The ecological security status of the land in this region not only affects the local ecological security of the Chengdu Plain region but also has significant implications for the ecological security of the middle and lower reaches of the Yangtze River. Therefore, gaining a comprehensive understanding of the LES status in the Chengdu Plain region is of immense theoretical and practical significance for the sustainable development of the Yangtze River Economic Belt. However, there is currently a scarcity of research specifically focused on land ecology in the evaluation of the ecological environment in the Chengdu Plain region, highlighting the need for more studies in this area. Although there has been growing attention to assessing the value of ecosystem services, further research is required to deepen our understanding in this field [27]. Studies on land use and land cover change have predominantly focused on topics such as urban expansion [28], agricultural production [29], and ecological conservation [30], with limited research specifically addressing the impacts on land ecology. It is evident that the Chengdu Plain region lacks a systematic LES evaluation system, which poses challenges in scientifically assessing the status and evolving trends of LES in the region [31–33]. Given this disparity, this study aims to address the gap by constructing a comprehensive set of LES indicators based on the PSR framework. These indicators will be used to evaluate the land ecological security status in the Chengdu Plain region, and their validity will be verified through the valuation of ecosystem services. The findings will contribute to a better understanding of the current state and evolving trends of LES in the region, providing a scientific basis for its sustainable development. Moreover, existing research predominantly focuses on ecological security management at the county level [26,31] or 1 km grid cells [28], which fails to meet the demand for more detailed management in

the medium-scale research area of the Chengdu Plain region. To address this issue, the research scale in this study has been narrowed down, and a grid cell size of 100 m has been identified as an optimal balance between detailed management requirements and data redundancy.

The objectives of this study are as follows: (1) construct an indicator system based on the PSR model and develop an evaluation model by employing a composite weighting method to determine the weights of the indicators; (2) utilize the evaluation model to assess the ecological security status of land resources in the Chengdu Plain region; (3) validate the credibility of the evaluation model by using the total value of ecosystem services as a validation measure; and (4) analyze the spatial distribution pattern of LES in the Chengdu Plain region.

## 2. Research Area and Methods

### 2.1. Study Area

The Chengdu Plain region, also known as the West Sichuan Plain, is located in the southwestern part of China and is the largest alluvial plain in the southwest region [34]. Spanning between longitude 103° E and 105° E and latitude 29.5° N and 32° N, it covers an approximate area of 18,000 square kilometers. Geographically, the Chengdu Plain region appears flat internally, but it actually encompasses various landforms such as mountains, hills, and plains due to the sloping geological structure [34]. The region has a subtropical humid, monsoon climate with distinct seasons and abundant precipitation, with an annual rainfall ranging from 1200 to 1600 mm [33].

The Chengdu Plain region is not only the economic center of Sichuan Province but also a densely populated area with a concentrated population and GDP [28]. In response to changes in domestic and international circumstances, promoting domestic circulation, serving the Yangtze River Economic Belt, and contributing to the Belt and Road Initiative, the concept of constructing the Chengdu–Chongqing Dual-city Economic Circle has been proposed, encompassing all cities in the Chengdu Plain region. This indicates that the Chengdu Plain region will continue to play a significant role as an economic center on a larger scale [35]. However, rapid economic development has resulted in significant human disturbances and impacts on the ecological environment. Therefore, it is crucial to assess the ecological security of land resources in the Chengdu Plain region to fulfill its responsibility as an ecological protection demonstration area in the upper reaches of the Yangtze River and as a central economic hub in the Midwest.

The Chengdu Plain region includes all districts and counties of Chengdu City, as well as selected districts and counties from Mianyang, Deyang, Meishan, Leshan, and Ya'an, totaling 43 units (As shown in Figure 1). This region possesses abundant natural and cultural resources, including majestic mountains, rivers on the plains, unique ecosystems, and a rich historical and cultural heritage. However, with the advancement of urbanization and the increase in economic activities, the land ecological security in the Chengdu Plain region is facing increasingly severe challenges. Therefore, there is an urgent need for a comprehensive assessment of the land ecological security in the Chengdu Plain region to protect and sustainably utilize these valuable natural and cultural resources.

In conclusion, as an integral part of the West Sichuan Plain, the Chengdu Plain region showcases rich geographical features and resource advantages. While achieving economic development, protecting land ecological security is crucial for ensuring sustainable development. Therefore, it is imperative to strengthen ecological protection and land resource management in the Chengdu Plain region to fulfill its mission and responsibility as an ecological protection demonstration area and a central economic hub in the Midwest.

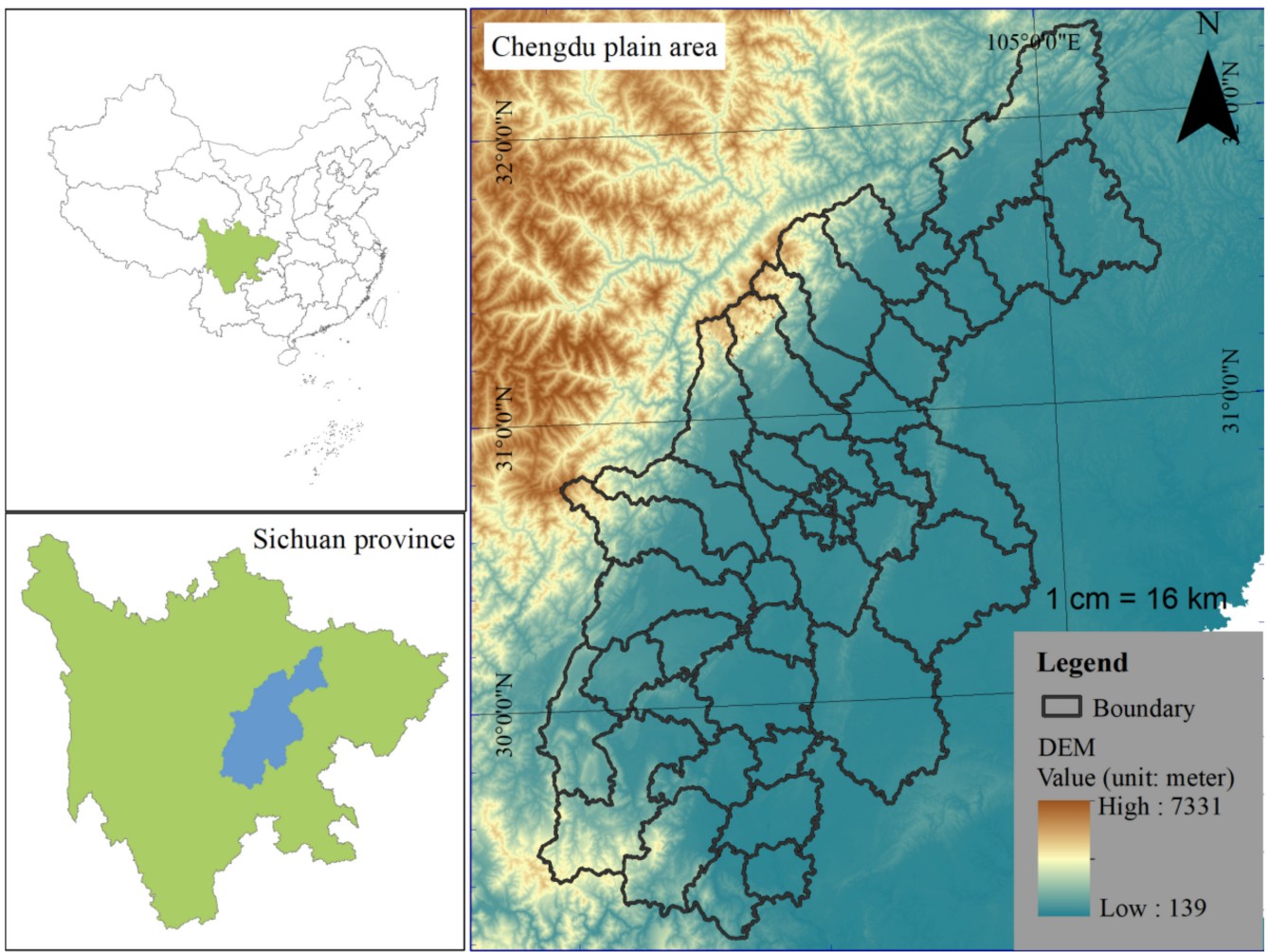

**Figure 1.** Location of the study area.

*2.2. Data Sources*

This study utilizes various datasets, including administrative boundary data, land use data, statistical data, and regional background value data, to comprehensively analyze the LES in the Chengdu Plain region. The details of these datasets, including their temporal years, spatial resolutions, formats, and sources, are summarized in Table 1.

**Table 1.** Research Data (2000–2020).

| Type | Data Name | Spatial Resolution | Format | Source |
|------|-----------|-------------------|--------|--------|
| Human factors | Land use | 30 m | Raster | Chinese Academy of Sciences, Resource and Environmental Science Data Center |
| | County-level administrative boundaries | \ | Vector | |
| | Annual GDP density | 1 km | Raster | |
| | Annual population density | 1 km | Raster | |
| | Annual nighttime lights [36] | 1 km | Raster | https://dataverse.harvard.edu/dataset.xhtml?persistentld=doi:10.7910/DVN/GIYGJU, accessed on 23 February 2023 |
| | Annual fertilizer application amount | County-level | Panel data | Statistical Yearbook |

**Table 1.** *Cont.*

| Type | Data Name | Spatial Resolution | Format | Source |
|---|---|---|---|---|
| Natural factors | Annual public financial expenditure | | | |
| | PM2.5 | | | Washington University in St. Louis, Atmospheric Composition Analysis Group |
| | Road | \ | Vector | Open Street Map |
| | Annual average temperature | 1 km | | National Earth System Science Data Center |
| | Annual average precipitation | 1 km | | |
| | Annual NPP | 1 km | | Chinese Academy of Sciences, |
| | Annual NDVI | 1 km | Raster | Resource and Environmental Science |
| | Soil texture | 1 km | | |
| | Ecosystem service value | 1 km | | Data Center |
| | DEM | 30 m | | Geographic Spatial Data Cloud |

(1) The land use data used in this study follows the "Classification Standard of Land Use" from the Third National Land Survey, ensuring consistency and compatibility with national land use classifications. Specifically, cultivated land primarily includes paddy fields and dryland; forest land mainly consists of forests, shrubs, and gardens; grassland includes alpine meadows and artificial lawns; water bodies encompass rivers, lakes, reservoirs, and other water-covered surfaces; construction land comprises urban and rural residential areas, industrial and mining areas, commercial and service areas, transportation areas, and other artificial surfaces; unused land refers to bare land surfaces not covered by vegetation or artificial structures.

(2) The county-level administrative boundary data used in this study were sourced from the "China Multi-Year County-Level Administrative Boundary Dataset" published by Xu Xinliang through the Resource and Environmental Science Data Registration and Publishing System. Specifically, the spatial distribution data of county-level administrative boundaries for the year 2018 were primarily selected for analysis.

(3) The spatial distribution of China's GDP at the kilometer grid level is derived from national county-level GDP statistical data. It takes into account the spatial interaction patterns between land use types, nighttime light intensity, population density data closely related to human activities, and GDP. The data is generated through spatial interpolation to create a 1 km × 1 km spatial grid dataset.

(4) In processing the spatial distribution of China's population at the kilometer grid level, the first step is to calculate the population distribution weights based on land use types, nighttime light intensity, and population density. These weights are then standardized, considering the influences from the three aspects mentioned above. Next, the total weights of each county-level administrative unit are calculated. Finally, by combining the population counts on the unit weight grid with the distribution map of total weights, the spatial distribution of the population is generated using raster spatial calculations.

(5) To assess the nighttime light intensity, annual nighttime light data is employed, which is derived from the Defense Meteorological Program-Operational Linescan System (DMSP-OLS) data. These data have been carefully corrected and improved by Wu Yizhen et al. [36]. and published in the journal *Transactions on Geoscience and Remote Sensing*. The dataset incorporates the "pseudoinvariant pixel" method to calibrate the DMSP-OLS data, considering the differences between the SUOMI National Polar-orbiting Partnership Visible Infrared Imaging Radiometer (SNPP-VIIRS) data and DMSP-OLS data.

(6) The data on annual fertilizer application and public fiscal expenditures were collected from the statistical yearbooks published by the Sichuan Provincial Statistics Bureau and the official websites of municipal governments. To achieve spatialization of panel data, in the case of fertilizer application data, this study averaged the fertilizer application amount within each county onto the arable land per square meter within the

county. Compared to the approach taken by Wang Juan et al. [37], who averaged it over the county area, the spatialization method used in this study for fertilizer application is more scientifically and logically justified. The panel data on public fiscal expenditures are compared based on the county area, reflecting the response intensity of county governments to land-related issues per square kilometer within their respective counties.

(7) The spatial distribution data of PM2.5 is sourced from the Atmospheric Composition Analysis Group at Washington University in St. Louis. We downloaded the data from their website, specifically opting for the highest precision data at a resolution of $0.01° * 0.01°$. The data was initially in .nc format and was subsequently converted to .tif format. We then aggregated and summarized the grid values to the respective administrative units for analysis.

(8) For analyzing the temperature patterns, the annual average temperature data is derived by averaging monthly temperature data to generate yearly data, providing a comprehensive understanding of the temperature conditions in the study area. By utilizing these datasets, this study aims to obtain accurate and reliable information for assessing the LES in the Chengdu Plain region.

(9) The spatial distribution dataset of the annual vegetation index (NDVI) in China is based on continuous time series of SPOT/VEGETATION NDVI satellite remote sensing data. It is generated using the maximum value synthesis method to create an annual vegetation index dataset. The annual NPP (Net Primary Productivity) data is derived from the MOD17A3 dataset, with a temporal resolution of 1 year and a spatial resolution of 500 m. It has been resampled to a 100 m resolution grid for analysis.

(10) The soil texture data in China is obtained from the compilation of soil profile data based on the 1:1,000,000 soil type map and the second soil survey. The soil texture classification is based on the content of sand, silt, and clay particles. The data is categorized into three major classes, sand, silt, and clay, and the content of different soil particles in each class is represented as a percentage.

(11) The spatial distribution dataset of ecosystem service values in China's terrestrial ecosystems is based on nationwide land ecosystem-type remote sensing classification data. It refers to the ecosystem service value equivalence factor method proposed by Xie Gaodi et al. [38]. The dataset estimates the values of four major categories and eleven types of ecosystem services for the years 2000, 2005, 2010, 2015, and 2020. The four major categories of ecosystem services include provisioning services (food production, raw materials production, water supply), regulating services (gas regulation, climate regulation, environmental purification, hydrological regulation), supporting services (soil retention, nutrient cycling, biodiversity), and cultural services (aesthetic landscapes). The dataset provides spatially distributed information on the values of these ecosystem services across different years.

First, all grid data were resampled to a spatial resolution of 100 m. The row and column numbers of the grids were aligned with the 2020 land use grid data as the reference. The projection coordinate system was set to WGS_1984_UTM_47N for consistency. For the land use data, it was reclassified into six major categories: cropland, forestland, grassland, water bodies, built-up land, and unused land. The soil texture data were classified into six categories based on the international standard for soil texture classification within the study area. Panel data, such as PM2.5 and public fiscal expenditures, were linked to the administrative units and converted into grid data with the same resolution, row and column numbers, and projection coordinates. Subsequently, the data were processed according to the evaluation indicators set by the Pressure-State-Response (PSR) framework. The specific procedures are in Section 2.3.1.

### 2.3. Method

This study focuses on examining the ecological issues associated with urban expansion from 2000 to 2020. A systematic approach is employed to achieve this, encompassing the establishment of an index system, determination of index weights, construction of an

evaluation model, model validation, and analysis of spatial patterns. By adopting this approach, the aim is to assess the current status and temporal trends of LES at the regional level. The research roadmap outlining the steps involved in this study is depicted in Figure 2.

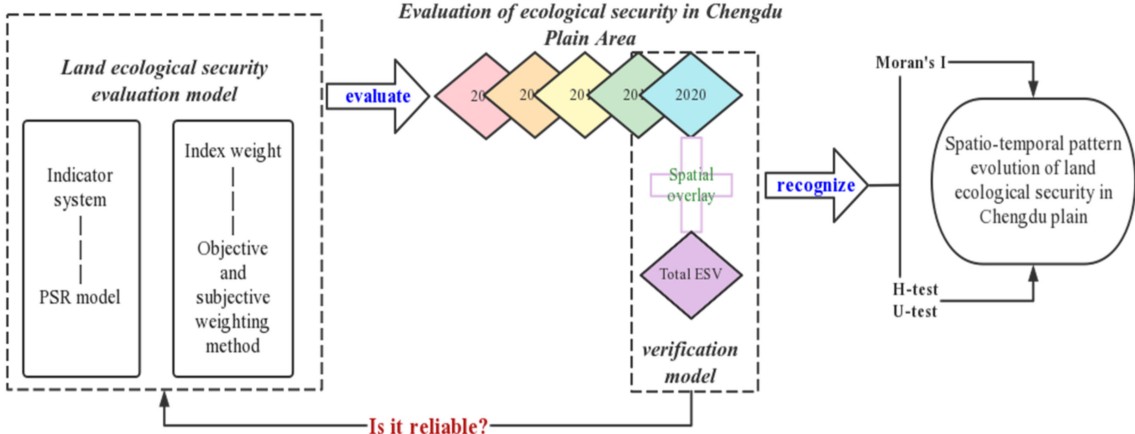

**Figure 2.** Technical route.

2.3.1. Construction of Ecological Security Evaluation System

The PSR model, initially proposed by the United Nations Economic Cooperation and Development Organization in 1990, is adopted as the theoretical framework. The PSR model provides a comprehensive approach to evaluateevaluating ecological security by considering the interplay between human activities and ecosystems in terms of pressure, state, and response [39,40]. By incorporating these three dimensions, the model effectively captures the complex relationships within ecosystems. Due to its robustness and applicability, the PSR model has been extensively utilized in research related to environmental sustainability and comprehensive assessment of ecological environments [41].

In this study, the Pressurepressure dimension mainly focuses on human activities' impacts on the land ecosystem. Two aspects are considered, including pollution generated by human emissions and the demand for human survival and development. The fertilizer and pesticide usage data are collected from the statistical yearbook for each county and compared with the arable land area to obtain the fertilizer and pesticide consumption per unit area to quantify pollution emissions. Pollution caused by vehicle exhaust and oil leakage on roads is quantified using Euclidean distance, and production and domestic waste and wastewater pollution are also measured using the same approach [42]. The pollution range from roads mainly concentrated within a 160 m radius, and the buffer zone within 100 m of secondary roads is classified as Level II, while the buffer zone within 100 m of primary roads is classified as Level I. The pollution range within built-up areas is divided into five levels based on the pollution extent. PM2.5 concentration, chosen as an indicator of pollution intensity, is consistent within each county's administrative boundary through table connections since PM2.5 data are panel data within each county and represent widespread air pollution.

The state dimension aims to assess the current status and resilience of the land ecosystem. In this study, the land ecosystem's current status is evaluated from four dimensions: climate, soil, vegetation, and landscape. The baseline values of the land ecosystem are represented by annual average temperature, annual precipitation, soil texture, and Normalized Difference Vegetation Index (NDVI). The regional water yield is considered as an implication within annual precipitation and topographic factors, while the soil stability factor is determined by combining topographic factors, soil factors, and land use conditions. To avoid duplicate calculations of topographic and water yield factors, they are not listed separately in the study. The soil stability factor and landscape fragmentation index are

used to measure the land ecosystem's resilience. The specific calculation formulas for these indicators are as follows:

(1)　**Soil stability**

The soil stability assessment method proposed by Zhao et al. (2020) is employed in this study to quantify soil stability in the Chengdu Plain region, utilizing ArcGIS software (ArcGIS Pro 3.0.1) [43]. ArcGIS Pro, a powerful geospatial analysis tool, is used to reclassify the slope, aspect, and land use types in the Chengdu Plain region according to the categories specified in Table 2. After reclassification, a weighted summation approach is applied, considering the assigned weights for slope, aspect, and land use types. The weighted summation combines these factors to generate the final soil stability map for the Chengdu Plain region.

**Table 2.** Grading scale for soil stability assessment indicators.

| Weighting | Type | Levels | Score |
|---|---|---|---|
| 0.3 | Slope | <5° | 10 |
| | | 5°–8° | 8 |
| | | 8°–15° | 7 |
| | | 15°–25° | 5 |
| | | 25°–35° | 3 |
| | | >35° | 1 |
| 0.3 | Aspect | −1 | 5 |
| | | 0–90 | 10 |
| | | 90–270 | 1 |
| | | 270–360 | 10 |
| 0.4 | Land use | Woodland | 10 |
| | | Grassland | 8 |
| | | Water area | 6 |
| | | Arable land | 2 |
| | | Building sites | 4 |

(2)　**Landscape Diversity Index**

Land use types are the fundamental units that constitute landscape ecosystems, and the stability of landscape structure and functionality is of great significance for the safety of land ecosystems [44]. The diversity and fragmentation of landscape structure constrain various ecological processes and impact the stability and biodiversity of regional ecosystems [44]. Landscape diversity refers to the diversity of different landscape types within a specific spatial extent. Drawing on studies by Nelson Katherine [45], Yu H. [46], and others, this study considers the use of Shannon's diversity index to characterize the landscape diversity in the Chengdu Plain region as relatively reasonable. The calculation formula for Shannon's diversity index is as follows:

$$SHDI = -\sum_{i=1}^{n} p_i ln_{p_i} \tag{1}$$

where *SHDI* is the Landscape Diversity Index, *n* represents the total number of landscape types in each grid, and $P_i$ represents the proportion of area in each grid occupied by landscape type *i*.

The response dimension focuses on the ability of nature or humans to address ecological problems. This framework characterizes the ability of human and environmental entities to address ecological issues. Per capita fiscal expenditure reflects the maximum response capacity of the government to regional environmental issues. A higher per capita fiscal expenditure indicates a greater response capacity of the government in dealing with ecological and environmental problems. In contrast, the Regional Development Index is a negative indicator in terms of response capacity, reflecting the residents' awareness of land protection. A higher Regional Development Index indicates stronger human-induced dam-

age and weaker conservation efforts, leading to a higher likelihood of ecological problems. The specific calculation method is as follows:

(1)    **Regional Development Index**

According to Liu et al. (1996) proposal, a comprehensive land use index and a land use model expression are used to assign values to different land types (Table 3), enabling the quantification of the degree of land use [47].

**Table 3.** Table of land use classification indices.

| Grading Index | 1 | 2 | 3 | 4 |
|---|---|---|---|---|
| Type of land use | Unused land | Woodland, water bodies, grassland | Arable land | Building sites |

The Regional Development Index is calculated using the following formula:

$$L = \sum_{i=1}^{n} A_i \times C_i \tag{2}$$

where $L$ is the composite index of regional development, and $A_i$ represents the percentage of area occupied by class $i$ of the total area.

In addition to the human response to environmental problems, the environment itself has a certain capacity to regulate, with areas of higher net primary productivity having a greater capacity to regenerate areas of ecological damage; the more resilient the ecosystem, the greater the capacity to withstand and repair environmental damage.

(2)    **Ecosystem resilience**

This paper uses the following formula to measure ecosystem resilience based on a reading of the relevant literature [48–51]:

$$ECO_{RES} = D \sum_{i=1}^{n} S_i P_i \tag{3}$$

where $ECO_{RES}$ is the ecosystem resilience, $D$ is the landscape diversity index, Landscape Diversity Index, $S_i$ is the percentage of the area of the $i$ landscape type, and $P_i$ is the resilience score of the $i$ landscape type. The resilience score is mainly determined by primary productivity, vegetation cover, or expert scoring. In this paper, the score is based on land cover type with reference to relevant studies in the Chengdu Plain region [52–54]. The results are shown in Table 4.

**Table 4.** Ecological resilience of the Chengdu Plain region $P_i$ score.

| Classification | Type of Land Use | Score | Description |
|---|---|---|---|
| 1 | Woodland | 95 | Woodlands play a decisive role in maintaining ecosystem resilience. |
| 2 | Grassland | 67 | Grassland is maintained and managed to increase the resilience limits of the area. |
| 3 | Water bodies | 59 | This category is of greater significance for maintaining the resilience of ecosystems and must be intensively managed and maintained. |
| 4 | Arable land | 37 | This category is of some significance for maintaining ecosystem resilience and must be used with care. |
| 5 | Building sites | 21 | This category is less significant for maintaining ecosystem resilience. |
| 6 | Unused land | 8 | This category contributes very little to ecological resilience and is even very degraded. |

Note: The data in the table are corrected from the relevant results of Zuo (2022) [54].

2.3.2. Normalization of Indicators

As the inconsistency of units between the selected evaluation indicators leads to no comparability between different types of data, it is necessary to normalize the various types of data. There are many ways to normalize data, and the article selects four data normalization methods based on the characteristics of the indicators with reference to numerous research results, with the following formulas:

**(1) Positive indicators** include average annual temperature, average annual precipitation, soil stability, NPP, ecosystem resilience, and average land financial input, calculated as follows:

$$X'_{ij} = \frac{x_{ij} - min(x_{ij})}{max(x_{ij}) - min(x_{ij})} \tag{4}$$

**(2) Negative indicators** include population density, GDP density, nighttime lighting, and Landscape Diversity Index, calculated as follows:

$$X'_{ij} = \frac{max(x_{ij}) - x_{ij}}{max(x_{ij}) - min(x_{ij})} \tag{5}$$

**(3) Intermediate indicators** include fertilizer load and regional development index, of which fertilizer load is in accordance with the "Fertilizer Application Technical Guidance Program for Major Crops in Sichuan Province 2018–2020", which states that the fertilizer application amount per square kilometer of land should be 20 kg, while the ecological response is strongest when most of the Regional Development Index is forest land, grassland, and water, and belongs to the intermediate type indicators specifically calculated as follows:

$$X'_{ij} = 1 - \frac{|x_{ij} - x_{best}|}{max|x_{ij} - x_{best}|} \tag{6}$$

**(4) The interval indicators** include the distance from the built-up area, PM 2.5 concentration, and NDVI, of which the pressure level does not change significantly within 100 m from the built-up area, so the interval is [0, 100]; PM 2.5 concentration is considered to be below 10 in accordance with international standards to meet the quality requirements; NDVI value is best when it is at [0.3, 0.5] for vegetation growth, and the specific calculation formula is as follows:

$$\begin{aligned} X'_{ij} &= \frac{a - x_i}{M}, (x_i < a) \\ X'_{ij} &= 1, (x_i \in [a, b]) \\ X'_{ij} &= \frac{x_i - b}{M}, (x_i > b) \end{aligned} \tag{7}$$

where $X'_{ij}$ represents the indicator for the *ij* cell after normalization; $X_{ij}$ represents the original value of the *ij* indicator for the cell; $max(x_{ij})$ is the maximum value of the *j* indicator for the *i* cell and $min(x_{ij})$ is the minimum value of the *j* indicator for the *i* cell; $x_{best}$ represents the best value for indicator *j*; and *a* and *b* represent the left and right interval endpoints of the best interval.

2.3.3. Weight Determination Using Combined Weighting Method

Both subjective weighting methods and objective weighting methods have inherent limitations that cannot be resolved through their own improvement. However, many researchers have found that combining subjective and objective approaches to determine weights can achieve a better balance between the two, to some extent avoiding the problem of excessive subjectivity and inadequate representation of data entropy [23]. Based on this, the weight determination in this study also adopts the combined weighting method, which combines subjective and objective approaches to determine the weights. In terms of subjective weight design, this study employs the expert rating method. Ten experts in the field of ecology were selected to assess the relative importance of paired indicators on a scale from 1–9. The individual judgments were then integrated into a final judgment matrix using geometric means. A consistency test was conducted on the judgment matrix, and the

consistency ratio was calculated to be 1.5743, rejecting the hypothesis of inconsistency in the judgment matrix. This process yielded the subjective weight values for each indicator. Regarding the objective weighting method, this study utilized the entropy weighting method, which represents the data variability. The specific calculation steps are as follows:

Firstly, *a* judgment matrix of *j* indicators for *i* evaluation areas is constructed. $A_{ij}$, firstly, the extreme difference method is used to standardize each indicator factor, where $a_{ij}$ is one of the evaluation indicators in the matrix.

Next, calculate the entropy value for each evaluation indicator:

$$H_i = K \times \sum_{i=1}^{n} P_{ij} \times In\left(P_{ij}\right) \tag{8}$$

$$P_{ij} = \frac{X'_{ij}}{\sum\limits_{i=1}^{n} X'_{ij}} \tag{9}$$

where, $(H_i \geq 0, K < -\frac{1}{ln\,n})$ and if $x_{ij} = 0$, then make $ln(P_{ij}) = 0$, then $H_i = 0$. Finally, the indicator weights are calculated:

$$W_i = \frac{1 - H_i}{n - \sum\limits_{i=1}^{n} H_i} \tag{10}$$

The weights determined by the expert scoring method and the weights determined by the entropy method were combined by the arithmetic equalization method, and the final indicator system is shown in Table 5.

**Table 5.** Ecological security evaluation indicator system.

| Target Layer | Guideline Layer | Indicator Layer | Weighting |
|---|---|---|---|
| Pressure | Pollution emissions | Pesticide and fertilizer loads | 0.0571 |
| | | Road pollution | 0.0143 |
| | | Pollution in built-up areas | 0.0143 |
| | | PM2.5 concentration | 0.0714 |
| | | Nighttime lighting data | 0.0143 |
| | Human interference | Population density | 0.0429 |
| | | GDP density | 0.0286 |
| Status | Climate | Average annual temperature | 0.0429 |
| | | Annual precipitation | 0.0429 |
| | Soil | Soil stability | 0.1 |
| | | Soil texture | 0.0571 |
| | Vegetation | NDVI | 0.0857 |
| | Landscape | Landscape fragmentation index | 0.0429 |
| Response | Humanity | Per capita financial expenditure | 0.0286 |
| | | Regional Development Index | 0.1286 |
| | Environment | NPP | 0.1143 |
| | | Ecosystem resilience | 0.1143 |

Based on the PSR model, the indicators were selected to assess the ecological security of the Chengdu Plain region using a weighted overlay in ArcGIS Pro. A comprehensive evaluation of pressure, state, and response indicators in the Chengdu Plain region was obtained. Additionally, the assessment of LES was derived through an accumulation model.

2.3.4. Validation Model for Total Ecosystem Service Value

Ecological services refer to the material products and intangible services that humans directly or indirectly obtain from the structure, processes, and functions of ecosystems, which are essential for sustaining life [55,56]. They mainly encompass

four categories: provisioning services, regulating services, supporting services, and cultural services. Ecosystem services have long been employed by numerous scholars to investigate the identification of ecological security patterns. Scholars have successively demonstrated the correlation and inherent mechanisms between ecological security and ecosystem service values [57–59]. The representativeness of service values in reflecting the ecological security status has been confirmed by a considerable body of scholarly research [60,61]. The assessment of ecosystem service values has become an important basis for ecologic environmental protection, ecologic functional zoning, environmental-economic accounting, and ecological compensation decision-making [62,63]. The total value of ecosystem services can, to some extent, reflect the health status of the structure and functions of an ecosystem. The higher the value of ecosystem services, the stronger the ecological carrying capacity and the greater the ecological security. Furthermore, LES is defined by the relatively high value and stable quantity of ecosystem services provided by land resources. In this study, the total value of ecosystem services was selected as the validation model for assessing the ecological security of the Chengdu Plain region.

The validation dataset for this study was obtained from Xu Xinliang's publication at the Data Center for Resources and Environmental Sciences of the Chinese Academy of Sciences. The dataset represents the spatial distribution of land ecosystem service values for the year 2020. Using the Fishnet tool in ArcGIS Pro, a 100 m interval grid was generated to sample the Land Ecological Security Index map and the Total Ecosystem Service Value map, resulting in a dataset of 242,392 records. Kendall Rank correlation analysis was performed using MATLAB, revealing a strong correlation coefficient of 0.4212, indicating a significant relationship between the two variables. However, while plotting the Land Ecological Security Index and the Ecosystem Service Value map, it was observed that their correlation patterns exhibited block-like distributions. Analysis of the Ecosystem Service Value dataset indicated that the values were concentrated within three threshold ranges, primarily around 0, 300, and 800. Moreover, certain areas exhibited extremely high values in the Total Ecosystem Service Value, making it challenging to achieve an ideal corresponding classification effect through dimensionless processing. In light of this, the study proceeded to classify the two datasets and calculate their correspondence through spatial overlay, considering the relationship between Ecosystem Service Value and Land Ecological Security levels. Specifically, areas with high Ecosystem Service Value generally exhibited higher ecological security, while areas with low Ecosystem Service Value tended to have weaker ecological security. By dividing the datasets into levels and performing spatial overlay calculations, a corresponding rate of 75.55% was achieved. However, due to the distinct three-segment distribution of the Ecosystem Service Value dataset, the critical values between segments were not clearly defined. The study determined the classification breakpoints as positions where the correspondence rate was maximized between the two concentrated distribution areas to mitigate subjective segmentation. After exploring all possible breakpoints, the optimal breakpoint was identified, resulting in a correspondence rate of 75.55%. The final results of the classification process are presented in Table 6.

**Table 6.** Corresponding grading of ecological security and ecosystem service values.

| Ecological Security Level | Number | Standards | Ecosystem Services Classification | Number | Standards |
|---|---|---|---|---|---|
| Unsafe | 1 | ≤0.45 | Low-Value Area | 1 | ≤100 |
| Safer | 2 | 0.45–0.6 | Median Zone | 2 | 100–500 |
| Safety | 3 | >0.6 | High-Value Area | 3 | >500 |

### 2.3.5. Study on LES Model in the Chengdu Plain Region

*Moran's I* is a statistical measure used to detect spatial autocorrelation, reflecting the degree of interaction between a particular attribute value within a region and its

surrounding areas. In this study, the global *Moran's I* was used to determine whether there is clustering among different dimensions of data in the Chengdu Plain region. Local *Moran's I* was employed to identify clustering areas within different dimensions. Additionally, the Mann–Whitney *U*-test and Kruskal–Wallis *H*-test were utilized to examine whether there are statistically significant differences in ecological security among different topographic areas, economic conditions, and urban–rural areas in the Chengdu Plain region.

According to the Chengdu Plain region geomorphological classification map, the region can be divided into plains and terraces, low hills, and mountains. For GDP zoning: The GDP density data was aggregated using the area tabulation tool to calculate the average values. The data were then sorted and divided into four categories based on a 25% interval. The top 25% of regions in terms of regional GDP were categorized as relatively developed areas, the next 25% were classified as relatively less-developed areas, and the bottom 25% were designated as economically backward areas, while other townships were classified as economically less-developed areas. For urban–rural zoning: The land use data were used to classify urban areas as urban construction land and categorize other land types as rural areas, thus creating an urban–rural division. For topographic zoning: The spatial distribution data of landforms at a scale of 1:1,000,000 in China was utilized. Plains and low-altitude plateaus were classified as plain areas, while high-altitude plateaus, farmland with elevation undulations, and low to moderate hills were classified as low hills and terrains. The remaining areas were designated as mountainous regions. The Mann–Whitney *U*-test and Kruskal–Wallis *H*-test were performed on this dataset, and when the *p*-value was less than 0.05, the test results were considered statistically significant.

## 3. Results and Analysis

### 3.1. Land Resource Ecological Security Status

In contrast to the weights obtained through the expert scoring method and entropy weighting method, the evaluation results obtained through the combined weighting method used in this study show significant discrepancies with the actual situation in the Chengdu Plain region. Among the three evaluation results, the evaluation result derived from the combined weighting method exhibits a high similarity to the distribution of the total value of ecosystem services. This finding validates the superiority of the combined weighting method employed in this study over other weighting methods.

The spatial overlay statistics between the ecological security evaluation results and the total value of ecosystem services in 2020 show an overlap rate of 75.55%. This indicates that the selected evaluation indicator system in this study can largely reflect the current status of LES in the Chengdu Plain region. The evaluation results are shown in Figure 3.

Based on this, the comprehensive indices of LES in the Chengdu Plain region during the period of 2000–2020 were obtained through the LES evaluation system. The indices are 0.5385, 0.5348, 0.5177, 0.5451, and 0.5545. All the comprehensive indices of ecological security are greater than 0.45, indicating that the LES in the Chengdu Plain region has been in a relatively safe state. The comprehensive indices of LES experienced a transition from a declining trend to a rising trend in 2010, with the lowest index of 0.5177. From 2000 to 2010, the LES index in the Chengdu Plain region shows a downward trend, with the most significant decline occurring from 2005 to 2010, with a decrease of 0.0171. From 2010 to 2020, the ecological security level gradually improves, with the fastest increase occurring from 2010 to 2015, with an increase of 0.0277. During the period of 2005–2015, there were drastic changes in land resource ecological security in the Chengdu Plain region, with a magnitude of change reaching 0.0448. These ten years coincide with the rapid economic development in the Chengdu Plain region and the increasing awareness of ecological protection. Therefore, the LES level in the Chengdu Plain region showed a deterioration due to earlier unsustainable development, followed by a rise in ecological security as measures were taken to protect the environment with the strengthening of ecological conservation awareness. Despite continued economic growth, the ecological security level showed an upward trend.

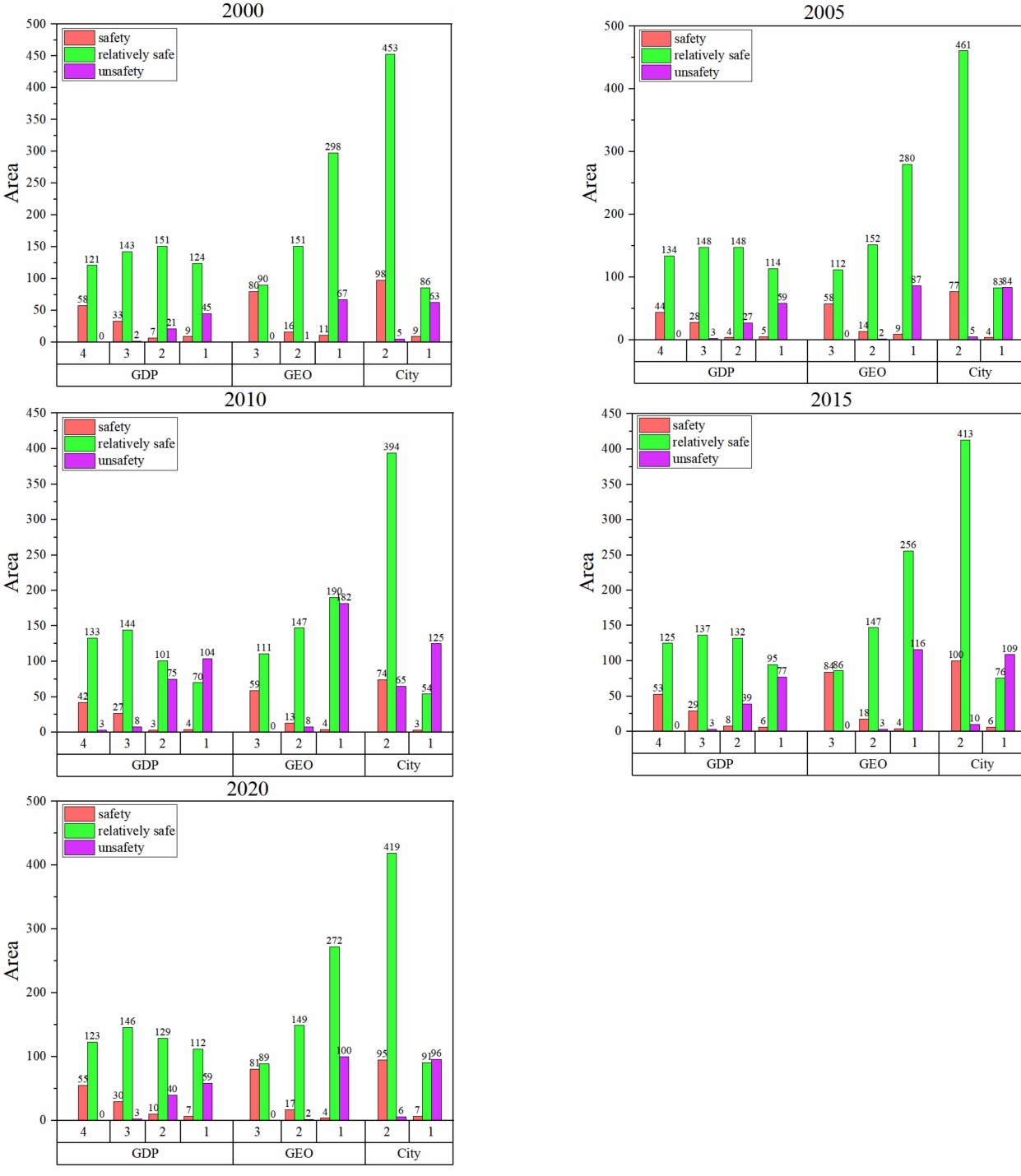

**Figure 3.** Bar chart depicting different land ecological security zones in different subregions for different years ("GDP" represents the economic development level zones, ranging from 4 to 1, indicating: highly developed economic areas, moderately developed economic areas, relatively underdeveloped economic areas, and economically backward areas. "GEO" represents the landform type zones, ranging from 3 to 1, indicating: mountainous region, low-hilly region, plain, and plateau. "City" represents the urban–rural division, with 2 representing rural areas and 1 representing urban areas).

### 3.1.1. Area Changes

In terms of area changes, there is an inverse trend between the unsafe and safe areas in the Chengdu Plain region (As Figure 4). The unsafe areas have been decreasing while

the safe areas have consistently increased. From 2000 to 2010, the total area of unsafe land resources in the Chengdu Plain region is 2634.1 km$^2$, 3515.65 km$^2$, and 9874.02 km$^2$. The unsafe areas exhibit rapid and continuous growth, with an annual increase of 724 km$^2$. The period between 2005 and 2010 experienced the highest growth rate, reaching 1271 km$^2$ per year. By 2010, the area of unsafe land resources in the Chengdu Plain region exceeds one-fourth of the entire plain, indicating concerning ecological security conditions. From 2010 to 2020, the area of unsafe areas decreases to 9874.02 km$^2$, 4930.34 km$^2$, and 3915.24 km$^2$, with an annual reduction rate of 596 km$^2$. In the past five years, the reduction rate has slowed down to 203 km$^2$ per year, but it has not yet reached the ecological security level of 2005. In contrast, the safe areas demonstrate an opposite trend, with respective areas of 25,730.97 km$^2$, 24,899.98 km$^2$, 17,506.23 km$^2$, 20,922.94 km$^2$, and 21,474.75 km$^2$. After 2010, the safe areas initially decreased and then began to increase. The area of ecological safe areas is 9033.80 km$^2$, 8986.24 km$^2$, 10,018.62 km$^2$, 11,545.59 km$^2$, and 12,008.81 km$^2$. Overall, the trend of ecological safe areas shows a steady increase throughout the 20-year period, with a small decrease observed only between 2000 and 2005. The study was conducted at a refined scale using a 100 m grid, which balanced the need for detailed management and minimized data redundancy.

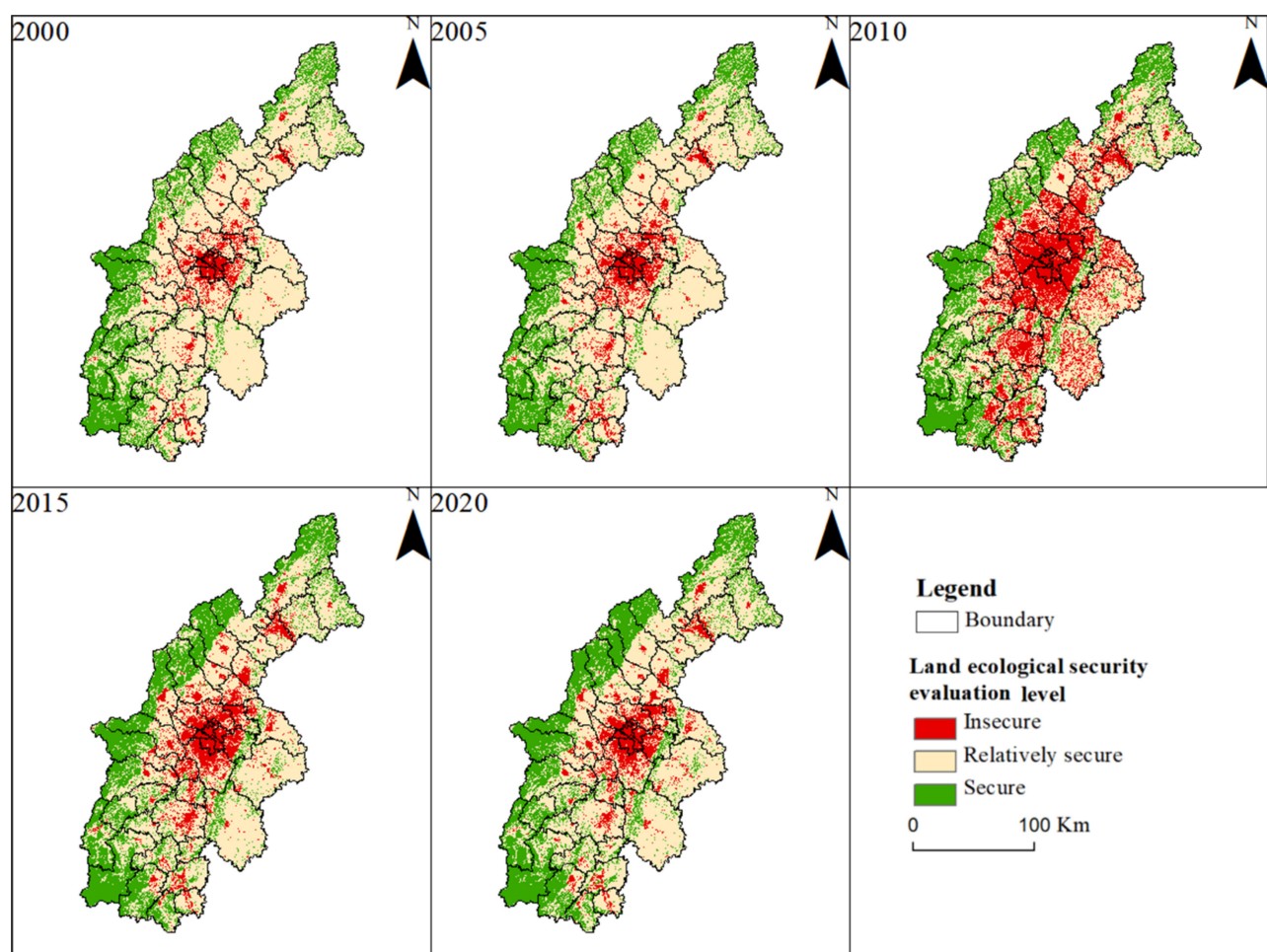

**Figure 4.** Results of land ecological safety evaluation at the grid scale in the Chengdu Plain region.

3.1.2. Changes in Spatial Distribution

Table 7 presents the results of the Mann–Whitney *U*-test and Kruskal–Wallis *H*-test for the land ecological security status in different regions of the Chengdu Plain. The results indicate significant regional variations in the land ecological security status of the Chengdu Plain. Figure 3 displays the areas of different land ecological security zones corresponding

to different subregions for the years 2000, 2005, 2010, 2015, and 2020. The specific details are as follows:

**Table 7.** Table of $p$ values for Mann–Whitney $U$-test and Kruskal–Wallis $H$-test for land ecological safety index.

| Year | Economic Status Zone | Topographical Zones | Urban and Rural |
|------|----------------------|---------------------|-----------------|
| 2000 | $4.7869 \times 10^{-47}$ | $6.7500 \times 10^{-73}$ | $2.0027 \times 10^{-36}$ |
| 2005 | $3.6883 \times 10^{-47}$ | $4.8699 \times 10^{-72}$ | $9.0976 \times 10^{-37}$ |
| 2010 | $2.3405 \times 10^{-43}$ | $5.5220 \times 10^{-79}$ | $1.0533 \times 10^{-35}$ |
| 2015 | $6.0647 \times 10^{-47}$ | $5.6837 \times 10^{-81}$ | $9.0141 \times 10^{-39}$ |
| 2020 | $1.2311 \times 10^{-42}$ | $4.4507 \times 10^{-77}$ | $1.2080 \times 10^{-35}$ |

Based on Table 7, there are significant differences in the land resource ecological security status among different economic level zones, landform zones, and urban–rural areas. GDP values, landform, and urban–rural structure profoundly influence the regional LES status. From Figure 3, the detailed differentiation among regions can be observed as follows:

(a) The LES status is related to the economic development level of the respective region. The majority of secure zones are located in the top 50% of the regional economic development level. Insecure zones mostly fall in the bottom 50% of the region's economic development level. The distribution of relatively secure zones does not strictly exhibit correlation, but the economically developed and economically backward areas have the largest areas of relatively secure zones.

(b) The spatial distribution of LES grade zones shows similarity with the spatial distribution of landform zones. Secure zones exhibit an increasing trend in distribution with the undulation and elevation of the landform in the Chengdu Plain region. Conversely, relatively secure and insecure zones show the opposite trend, with a decrease in distribution area with increased landform undulation and elevation. Insecure zones are mainly concentrated in the plain and plateau landform zones.

(c) The spatial distribution of LES zones and insecure zones shows similarity with the urban–rural spatial distribution. During the period from 2000 to 2020, secure and relatively secure zones were mainly distributed in rural areas, while insecure zones were primarily concentrated in urban areas. However, in 2010, although the area of insecure zones in rural areas increased, relatively secure and secure zones still dominated.

From the perspective of temporal dynamics, the main distribution areas of LES zones are in the Longquan Mountain and Longmen Mountain regions. In the foothill areas, the secure zone range experiences small-scale fluctuations influenced by economic development and urban expansion. The distribution of ecologically insecure zones roughly corresponds to the urban built-up area, and its range is broader than that of the built-up area, influenced by the level of economic development. Ecologically relatively secure zones, influenced by landforms, are primarily distributed in the plain and plateau areas with low hills and are also affected by urban expansion and economic development, leading to the conversion of certain areas into insecure zones. These factors collectively shape the spatial distribution pattern of relatively secure zones.

### 3.1.3. Changes at Township Scale

Examining the LES status of the Chengdu Plain region at the township scale reveals an increasing number of townships under ecological security (Figure 5). The period from 2005 to 2010 witnessed the highest increase in the number of ecologically unsafe townships, primarily radiating outward from the central urban area of Chengdu. After reaching its peak in 2010, the number of ecologically unsafe townships has been steadily decreasing. On the other hand, the number of townships under ecological security has continuously increased over the 21-year period, mainly concentrated in the Longmen Mountain area. However, from 2015 to 2020, some high-value ecological security areas also emerged in

the Longquan Mountain area, indicating that Chengdu has achieved certain success in ecological construction in the Longquan Mountain area in recent years. During the 21-year period, most townships in Ya'an and Leshan were in a state of LES, while the majority of townships and streets in Chengdu belonged to the category of land ecological insecurity, requiring significant attention and remedial measures.

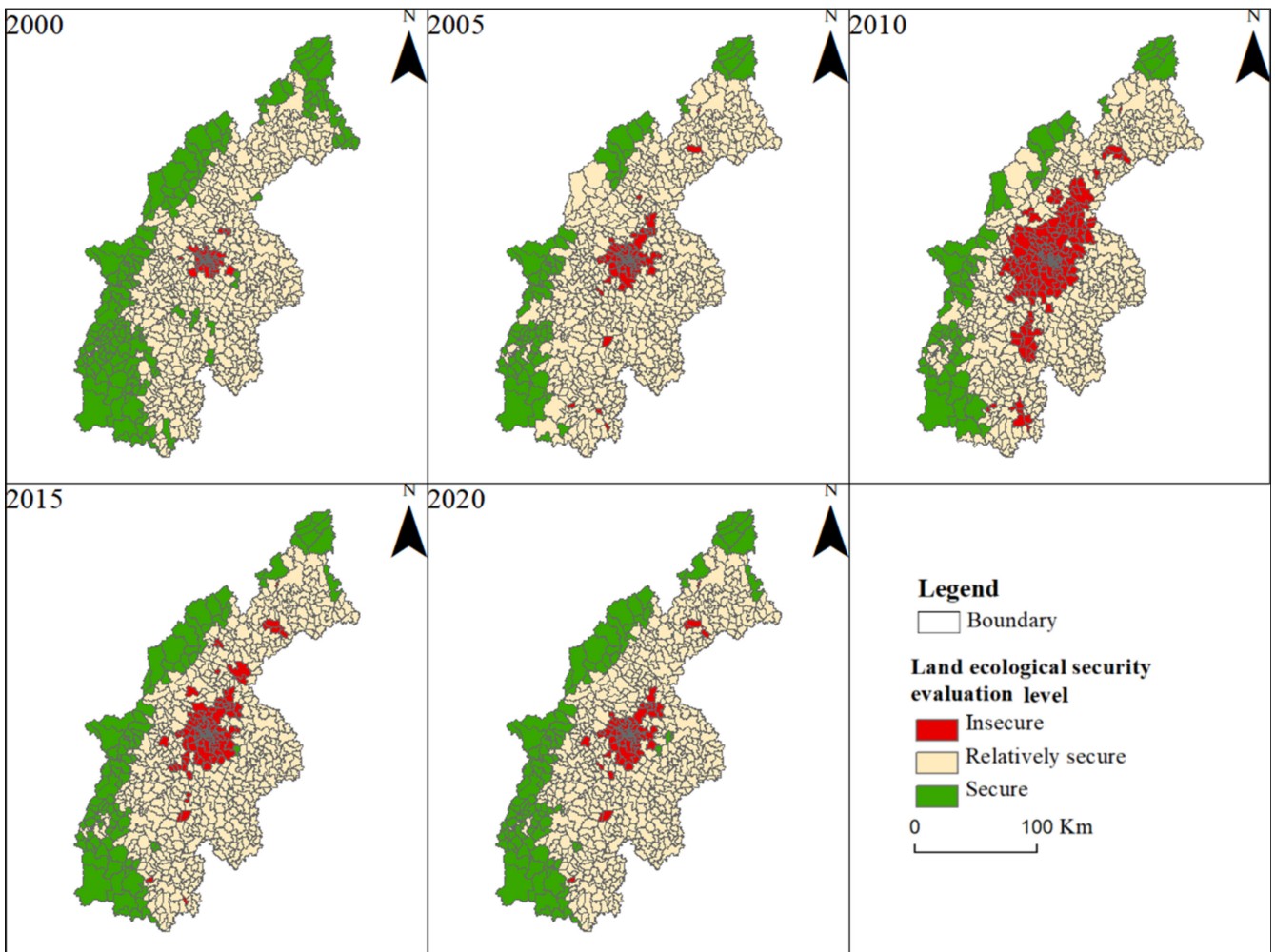

**Figure 5.** Results of the evaluation of LES at the township scale in the Chengdu Plain region.

*3.2. Spatial Clustering in Different Dimensions*

The global Moran indices of the three dimensions of pressure, state, and response for the five periods of land resource ecological security evaluation from 2000 to 2020 were all at the 99% confidence level (Table 8). As can be seen from Table 8, the Moran indices of the three dimensions for 2000–2020 are all above 0.7, and the Chengdu Plain region as a whole shows a significant spatial clustering of LES.

**Table 8.** Summary of the global Moran index.

|          | 2000   | 2005   | 2010   | 2015   | 2020   |
|----------|--------|--------|--------|--------|--------|
| Pressure | 0.8654 | 0.8632 | 0.8704 | 0.8343 | 0.8471 |
| Status   | 0.7273 | 0.7712 | 0.8001 | 0.7986 | 0.8192 |
| Response | 0.8375 | 0.8391 | 0.8411 | 0.8434 | 0.8442 |

The numerical values in the stress dimension indicate that higher values represent lower stress levels, while lower values indicate higher levels of stress. The results are

shown in Figure 6. The analysis of local *Moran's I* reveals that clusters with high ecological stress are primarily concentrated in the central urban area of Chengdu. From 2000 to 2015, as the economic development of townships in the surrounding counties progressed, high-stress clusters showed a spreading trend centered around Chengdu, extending in the north-south direction. There were even high-stress clusters observed in townships in the eastern parts of Meishan City and Leshan City. However, during the five-year period from 2015 to 2020, with the strengthening of ecological protection efforts and the emphasis on ecological conservation alongside economic development, the area of high-stress clusters in the Chengdu Plain region significantly reduced and was mainly confined to the densely populated and economically active central-urban area of Chengdu. The spatial distribution of clusters with low ecological stress remained relatively unchanged, primarily located in the Longmen Mountain area.

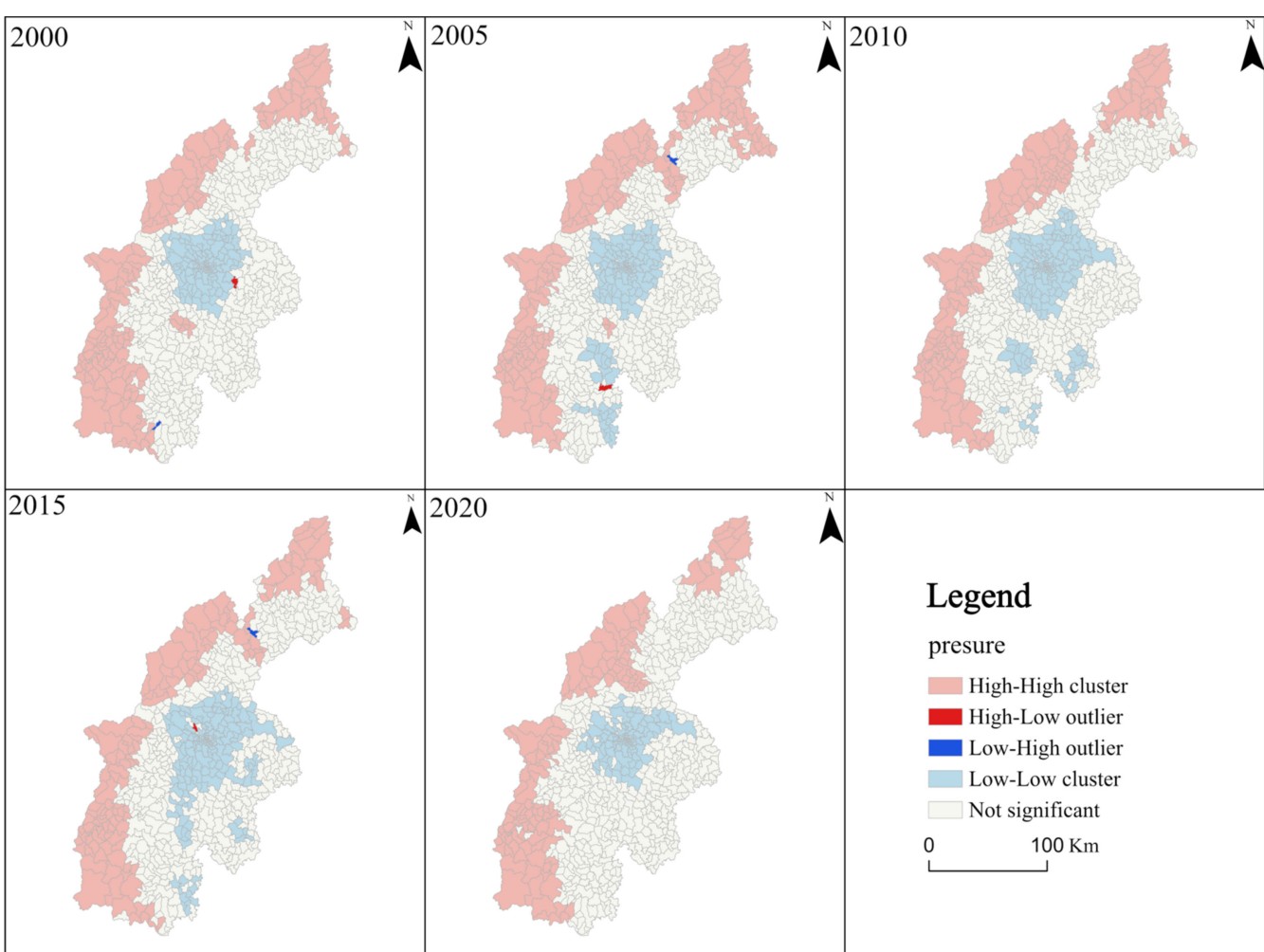

**Figure 6.** Local Moran index mapping of the pressure dimension in the Chengdu Plain region.

The numerical values in the status dimension indicate that higher values represent better ecological conditions, while lower values indicate poorer conditions. The results are shown in Figure 7. The clusters in the status dimension have undergone significant changes primarily due to alterations in land cover in the Chengdu Plain area. The baseline values of ecological status in the Chengdu Plain region show little variation. The main differences lie in the changes in soil stability and fragmentation resulting from changes in land use. From the graph, it can be observed that high-value clusters of ecological security are mainly located in the southern part of Longmen Mountain, encompassing the cities of Ya'an and Leshan. This region is predominantly hilly and mountainous, with abundant

forest resources. The rugged terrain is not conducive to large-scale economic activities. However, the favorable natural environment has contributed to the development of tourism in the area. With continuous economic growth, increased consumer spending power, and growing interest in tourism, the region has intensified its environmental protection efforts to maintain its attractiveness as a tourist destination. As a result, the number of high-value clusters in this area has increased from 2000 to 2020. Over time, some mountainous townships that are unsuitable for industrial development have also shifted their focus to tourism, emphasizing the development of ecological resources and environmental protection. As a result, high-value clusters have emerged in the northern and Longmen Mountain areas, making the Longmen Mountain region the main distribution area of high ecological status in the Chengdu Plain region. On the other hand, within the Chengdu City area, changes can be observed since 2000. The high-value clusters in the eastern part have gradually diminished, and by 2020, there are almost no high-value clusters remaining. In contrast, the low-value clusters in the central part have expanded and are trending southward. Considering Chengdu's urban development plan, the city has been advocating for eastward and southward expansion in recent years, resulting in changes in land use and conversion of some ecological land to construction land, which has had a significant impact on regional ecological security. The spatial distribution of low-value clusters has transitioned from dispersion to aggregation. In 2000, the low-value cluster distribution in the Chengdu Plain region was scattered, but over the 21-year period, it gradually aggregated towards the regional economic center, indicating that the factors influencing regional ecological security have gradually shifted towards urban development-induced changes.

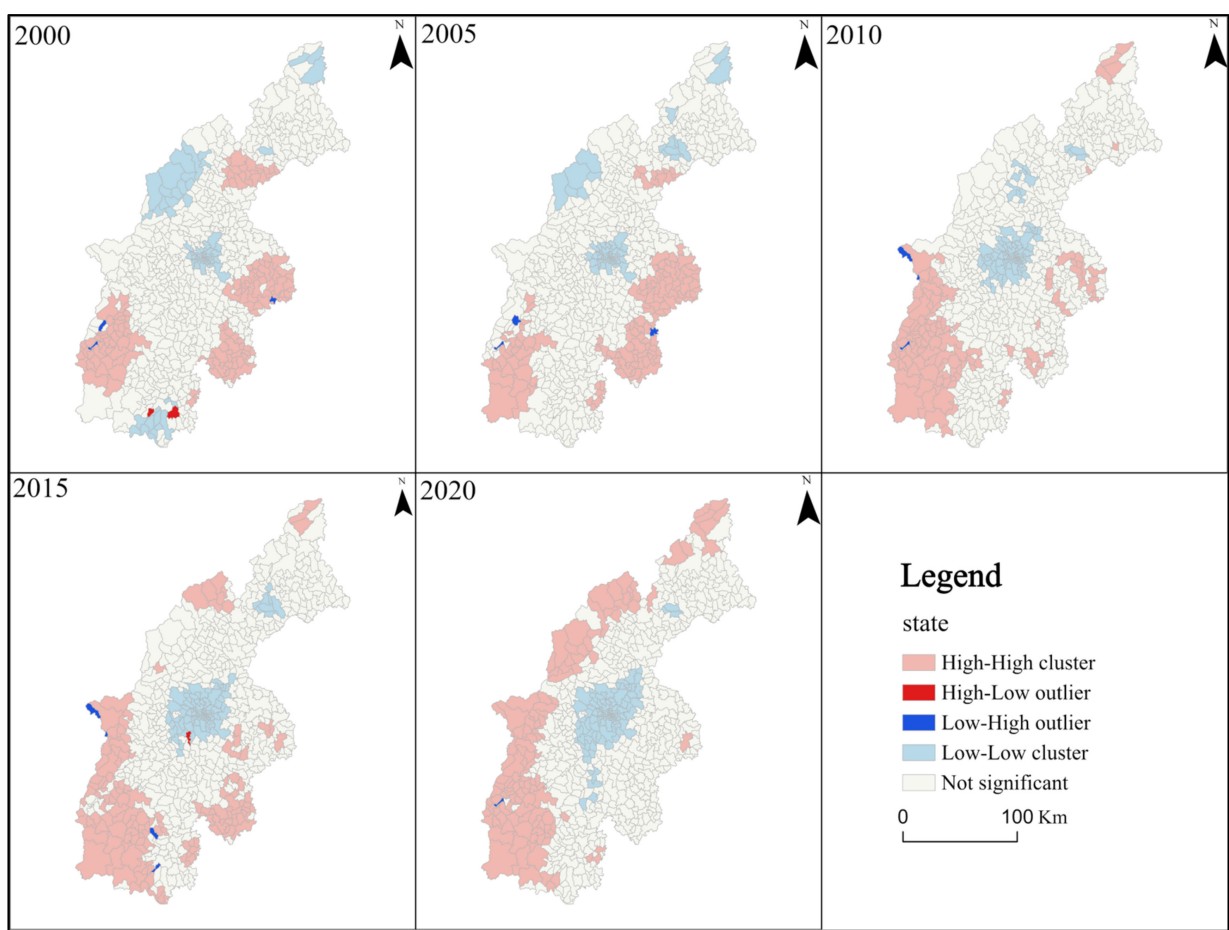

**Figure 7.** Local Moran index mapping for the state dimension in the Chengdu Plain region.

In the response dimension, larger numerical values indicate stronger regional response capacity, while smaller values indicate weaker response capacity. The results are shown in Figure 8. The distribution of regions with lower ecological response levels is relatively concentrated, primarily located in the five urban districts of Chengdu City, with a tendency to expand toward the surrounding areas. The ecological response level in the Chengdu Plain region remains relatively stable over the 21-year period. Regions with higher response levels are mainly found in the Longmen Mountain area, where the natural response capacity is stronger. Regions with lower response levels are concentrated within the Chengdu City area, indicating weaker natural and human-induced response capacities. The low-value cluster reached its maximum extent in 2015, and by 2020, the response level in Chengdu City had improved to some extent.

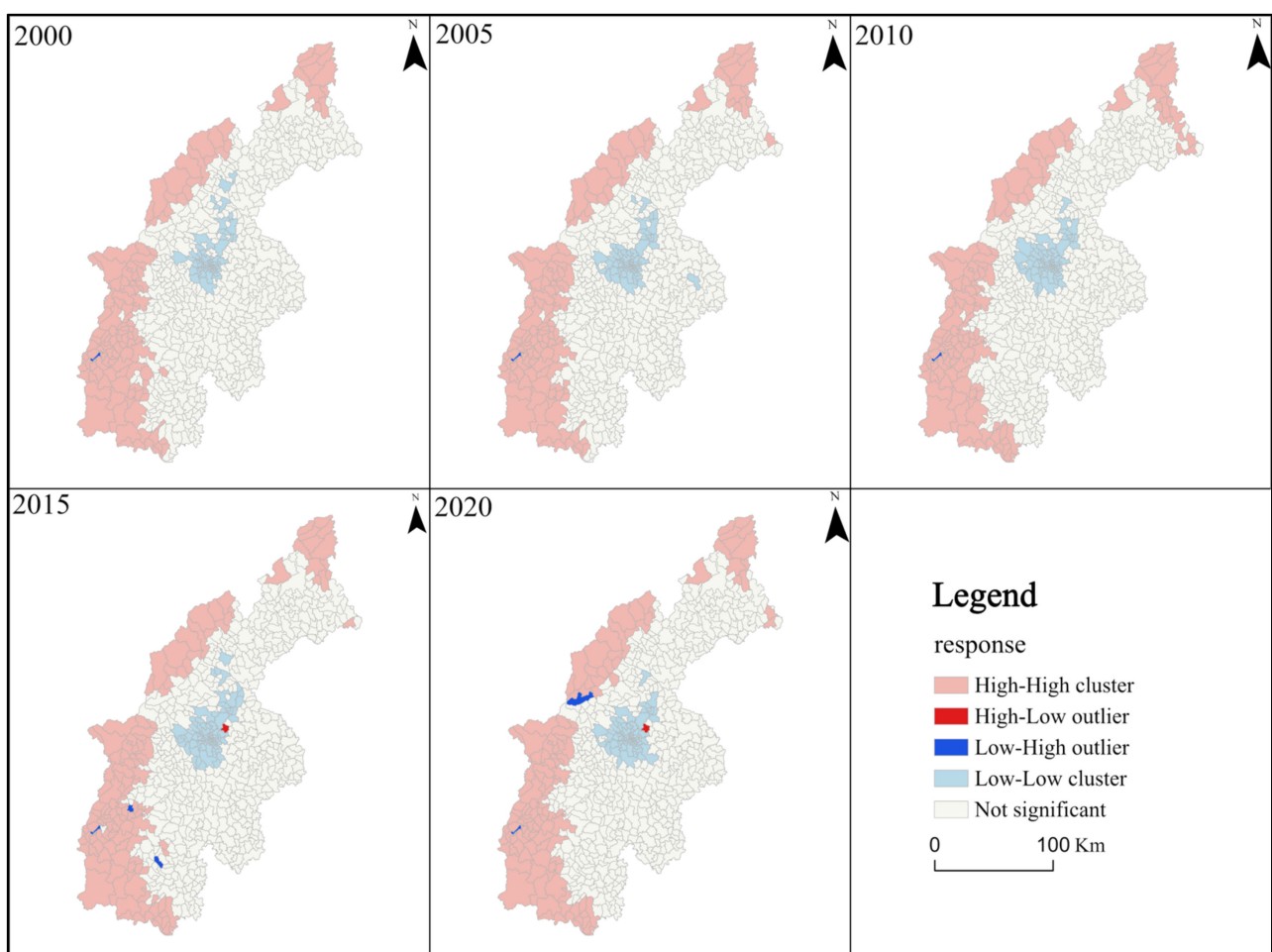

**Figure 8.** Local Moran index mapping of response dimensions in the Chengdu Plain region.

Table 9 presents the results of the Mann–Whitney *U*-test and Kruskal–Wallis *H*-test for the evaluation of LES in three dimensions in the Chengdu Plain region. The results indicate that the economic development, topography, and urban–rural structure have an impact on the ecological security of land resources in the various townships within the Chengdu Plain region.

**Table 9.** Table of *p*-values for Mann–Whitney *U*-test and Kruskal–Wallis *H*-test for the three dimensions.

| Year | Dimensionality | Economic Status Zone | Topographical Zones | Urban and Rural |
|------|----------------|----------------------|---------------------|-----------------|
| 2000 | Pressure | $3.0889 \times 10^{-33}$ | $2.8112 \times 10^{-68}$ | $1.2956 \times 10^{-22}$ |
| | Status | $9.4585 \times 10^{-29}$ | $1.7732 \times 10^{-17}$ | $4.6918 \times 10^{-29}$ |
| | Response | $8.4065 \times 10^{-47}$ | $1.4864 \times 10^{-79}$ | $3.4955 \times 10^{-39}$ |
| 2005 | Pressure | $1.3626 \times 10^{-27}$ | $2.5451 \times 10^{-55}$ | $1.5268 \times 10^{-21}$ |
| | Status | $1.4600 \times 10^{-33}$ | $1.5327 \times 10^{-22}$ | $2.8957 \times 10^{-35}$ |
| | Response | $1.2757 \times 10^{-46}$ | $1.4864 \times 10^{-79}$ | $2.0002 \times 10^{-41}$ |
| 2010 | Pressure | $1.1189 \times 10^{-22}$ | $2.8106 \times 10^{-52}$ | $1.1523 \times 10^{-21}$ |
| | Status | $8.9158 \times 10^{-39}$ | $1.3918 \times 10^{-64}$ | $1.3757 \times 10^{-40}$ |
| | Response | $1.0127 \times 10^{-50}$ | $1.1004 \times 10^{-85}$ | $1.2690 \times 10^{-48}$ |
| 2015 | Pressure | $4.3037 \times 10^{-26}$ | $5.5282 \times 10^{-60}$ | $6.5240 \times 10^{-20}$ |
| | Status | $1.5700 \times 10^{-39}$ | $1.1659 \times 10^{-68}$ | $7.3012 \times 10^{-46}$ |
| | Response | $4.3349 \times 10^{-51}$ | $1.9927 \times 10^{-80}$ | $3.1184 \times 10^{-50}$ |
| 2020 | Pressure | $1.0939 \times 10^{-29}$ | $1.1524 \times 10^{-59}$ | $8.7773 \times 10^{-30}$ |
| | Status | $3.5177 \times 10^{-38}$ | $5.3716 \times 10^{-75}$ | $7.5622 \times 10^{-43}$ |
| | Response | $1.3650 \times 10^{-49}$ | $1.5245 \times 10^{-77}$ | $1.5218 \times 10^{-50}$ |

Considering the local *Moran's I*, there are significant differences in the ecological security of land resources between economically developed areas and relatively developed townships in the Chengdu Plain region compared with economically backward and relatively backward regions. From a topographic perspective, there are significant differences in ecological security between mountainous terrain areas and relatively gentle plain plateaus and low hills. Mountainous terrain areas are often associated with safe zones, while plain plateaus and low hills correspond to relatively safe and unsafe regions. The distribution of relatively safe and unsafe areas within these topographic zones is largely influenced by economic levels and urban–rural distribution.

## 4. Discussion

In response to the significant uncertainty in the current assessment of land ecological security, this paper proposes a comprehensive evaluation model that combines the advantages of subjective and objective weighting methods. By balancing the expert knowledge in the subjective weighting method and the data characteristics in the objective weighting method, the shortcomings of a single weighting method, such as excessive subjectivity and inconsistencies with reality, are avoided. Researchers such as Li Peiwu [64] recognized the limitations of the Analytic Hierarchy Process (AHP) and entropy weight method in the ecological security assessment of Shenzhen and combined the two methods by assigning a preference coefficient of 0.5 to determine the combined weights, ultimately obtaining an ecological security assessment model for Shenzhen. Wei He [21] suggested that to avoid arbitrary, subjective weights and contradictions between objective weights and practical experience, it is necessary to organically combine AHP with the entropy weight method, ensuring a good balance between the two and achieving better alignment between indicator weights and reality. Yiran Wang [65] utilized principal component analysis and grey clustering methods to classify and evaluate the forest ecological security levels of 11 provinces (municipalities) in the Yangtze River Economic Belt and obtained favorable evaluation results. Xinchang Zhang [66] combined AHP with the improved grey relational TOPSIS method to accurately evaluate the ecological security status of land in mining cities. The studies conducted by these scholars have confirmed that the combination of subjective and objective weighting methods can yield evaluation results that are more consistent with the actual situation. Therefore, based on the conventional AHP method, this paper integrates the knowledge and experience of multiple experts to obtain subjective weights and combines them with the entropy weight method that reflects data characteristics, ultimately achieving a unified set of indicator weights. In the combined subjective-objective method of composite weighting, in order to avoid significant deviations in the weights of individual

indicators caused by excessive entropy values of certain data in the objective weighting method, data cleaning is required. Additionally, to prevent data redundancy resulting from a large volume of data, data lightweight is necessary. The complexity of data processing is further heightened by multiple steps, such as normalization and standardization of indicators. Furthermore, the determination of trade-offs between weights obtained from subjective and objective sources requires more reliable data and accurate expert opinions, as well as careful handling of different expert perspectives and weight considerations.

This paper introduces, for the first time, a verification model that utilizes the widely recognized and fundamental measure of the Ecological System Service (ESS) value, which is commonly used as a basis for ecological conservation, to validate the level of Land Ecological Security (LES). The relationship between the ESS value based on human demands and the evaluation of ecological security was initially proposed by Zuo Wei [67]. Subsequently, researchers such as He Lin [68], Li Can [69], Wang Zhengwei [70], and Wang Yun [71] adjusted the ecological security pattern based on ESS value, confirming the scientific correlation between ESS value and ecological security. In comparison to the approach taken by Cheng et al. [1], which validates the model through corresponding individual dimensions, using ESS value as the ultimate validation model for assessing land resource ecological security provides a more intuitive reflection of the overall reliability of the model. Furthermore, compared to the methods employed by Zhao Yuting [24], Liu Chenli [26], and others, who use multidimensional indicators for model verification, adopting a single comprehensive indicator allows for a more comprehensive consideration of the scope and provides clearer and more intuitive verification results. Although there is a corresponding relationship between the value of ecosystem services and land ecological security, it is important to note that this relationship is not strictly linear. This is fully reflected in the maximum validation accuracy of up to 75.55%. While some of the errors in between can be attributed to inconsistencies in the original data resolution, there are also differences in the definitions of the two concepts. For example, in the case of unused land, the ecosystem service value may be calculated as zero, but it does not necessarily indicate an unsafe condition in terms of land ecological security. This is also related to human demands imposed on that land. Fragile land ecosystems, which receive greater attention and protection and have lower human demands, may exhibit convergent changes in land ecological security influenced by the surrounding environment. This is particularly evident in the unused land on the western side of Longmen Mountain.

Thinking about the relationship between ecosystem service value and land ecological security reveals the interactive balance between economic development and ecological protection. The potential conflict between ecological security and economic development stems from the competitive demand for natural resources. Economic activity often puts pressure on ecosystems, leading to degradation and a decline in ecological security. However, ecological security is also crucial for maintaining long-term economic development. Environmental degradation can have adverse effects on various economic sectors such as agriculture, tourism, and public health. Thus, achieving a balance between the two requires dealing with potential conflicts and exploring synergies, which involve the implementation of policies and regulations that promote sustainable resource management and conservation measures and incorporate ecological considerations into economic decision-making processes. By adopting an ecosystem-based approach, policymakers can identify win-win solutions while strengthening ecological security and supporting economic growth.

Furthermore, the spatial clustering of the evaluation scores for the pressure, state, and response dimensions was effectively demonstrated using global *Moran's I* and local *Moran's I*. The spatial differentiation of economic development level, topography, and urban–rural structure was analyzed using the Mann–Whitney *U*-test and Kruskal–Wallis *H*-test. Due to the influence of topography, cities in the Chengdu Plain area are mainly distributed in the plain regions, resulting in the concentration of industries and population in these areas. However, when the concentration of population and industries becomes excessive and construction land encroaches upon ecological land, the regional

ecological pressure becomes too high, leading to ecological insecurity. In the low-hilly regions, cities are unable to concentrate in a contiguous manner due to topographical constraints, resulting in an intermixing of urban and rural areas. In this case, the distribution of population and industries is appropriately balanced, leading to a certain equilibrium between ecological pressure and carrying capacity, resulting in a relatively secure ecological state. In mountainous areas, the significant terrain variations make the region unsuitable for human production activities. Therefore, although the ecological state in these areas is generally average, the low human-induced pressure contributes to an ecological security status.

The study also found that the Chengdu Plain region as a whole is in a relatively secure ecological state. However, within this relatively secure state, there was a turning point around 2010, with a trend of initial deterioration followed by improvement in the land's ecological security. The main cause of this deterioration and subsequent improvement is the change in the number of areas categorized as relatively secure and ecologically insecure. During the period from 2000 to 2010, extensive conversion of farmland, grassland, and woodland into construction land occurred due to the need for economic development. Additionally, the presence of unused land resulting from improper construction planning severely undermined regional land ecological security. Consequently, a significant number of areas classified as relatively secure experienced a shift to ecologically insecure areas, particularly during the period of rapid economic development from 2005 to 2010 in the Chengdu Plain. These changes were primarily concentrated around urban land in various districts and counties, as confirmed by the Mann–Whitney U-test and Kruskal–Wallis *H*-test. From 2010 to 2020, there was an increased emphasis on ecological environment protection in national policies, leading to greater attention from various sectors. Furthermore, the construction of ecological conservation zones in the upper reaches of the Yangtze River, along with the shift from extensive to intensive economic development in the Chengdu Plain region, contributed to a slowdown in urban land expansion, the implementation of land restoration projects (such as returning farmland to forest and grass), and the development of eco-friendly cities. As a result, the area of ecological land in the Chengdu Plain gradually increased. The progress of ecological restoration projects led to the transformation of some ecologically insecure areas into relatively secure areas and even into ecologically safe areas.

In summary, this study provides a comprehensive model for assessing land ecological security, successfully evaluating the land ecological security level in the Chengdu Plain region. However, research needs to be further improved and refined. Although this study constructs an indicator system based on the PSR model, taking into account various dimensions and aspects, the subjective nature of indicator selection introduces individual characteristics. For instance, GDP values have been used as indicators in the pressure dimension [72], as well as in the response dimension by considering industrial segmentation [73,74], and even as indicators in the state dimension by normalizing them per capita [72,75]. NDVI values, which reflect regional vegetation coverage, are commonly used for evaluating the state dimension [76], and some have further calculated forest coverage or green space coverage as indicators in the response dimension [72]. Furthermore, the completeness of regional data can also affect indicator selection. In the process of selecting data for the pressure and response dimensions, the statistical criteria and indicators vary across different cities and years. Consequently, energy consumption data, pollution control investment data, and education levels, among others, were unavailable, and alternative indicators had to be chosen to indirectly reflect such data. Thus, the 17 selected indicators may not comprehensively represent the ecological security status of the Chengdu Plain region, highlighting the need for further research in indicator system selection and the incorporation of logical and methodological approaches from operations research and management science to enhance objectivity.

Inconsistent spatial resolution impacts the final validation accuracy. When evaluating the ecological security of land resources in the Chengdu Plain region, this study considered the small-scale nature of the region and found that using 1-km grid data may not meet the

requirements for fine-scale management of land resources. Therefore, multiple resolutions were explored, and ultimately, 100 m grid data were chosen for the study. However, the validation data used in this study were obtained from the Data Center for Resources and Environmental Sciences of the Chinese Academy of Sciences, and the ecosystem service value data provided on their website had a resolution of 1 km. Although bilinear interpolation was employed to downscale the data, the precision still falls short of the validation data's accuracy. As a result, in regions with small aggregation ranges of ecological security evaluation results or transitional zones between two evaluation results, the downscaled ecosystem service value data fails to accurately represent the data details, leading to misclassifications in certain data details.

## 5. Conclusions

This study conducted an evaluation of the LES status in the Chengdu Plain region using the PSR framework and a combination weighting method. The ecosystem service total value was selected as the validation data to assess the reliability of the evaluation model. Furthermore, global *Moran's I* and local *Moran's I* were employed to examine the spatial autocorrelation of each dimension and the clustering patterns of high and low values. Additionally, the Mann–Whitney *U*-test and Kruskal–Wallis *H*-test were utilized to assess significant differences among different regions. The following conclusions were drawn.

(1) The weighting method employed in this study accurately reflects the contribution relationship between LES indicators in the Chengdu Plain region and the actual ecological security status. The resulting LES evaluation model can reliably represent the current ecological security status of the Chengdu Plain region, providing valuable references for relevant authorities in environmental protection and sustainable land development. Moreover, this study, for the first time, proposed the use of Total Ecosystem Service Value as a validation model for evaluation results based on the connotation of productivity in the definition of LES. Compared to multidimensional validation models, this validation method comprehensively and intuitively reflects the reliability of evaluation results.

(2) The overall level of LES in the Chengdu Plain region is relatively safe. The composite safety index exhibits a decreasing trend followed by an increasing trend, with 2010 as the turning point. The areas of unsafe regions and relatively safe regions show an inverse relationship, with the area of unsafe regions reaching its peak in 2010 while the area of ecological security regions steadily increased.

(3) The spatial pattern of LES in the Chengdu Plain region exhibits significant spatial clustering and zoning. Overall, the insecurity of land ecology in the Chengdu Plain region is mainly attributed to human factors, and its spatial distribution closely corresponds to areas with frequent human activities. The spatial clustering of the pressure, state, and response dimensions is strongly influenced by the level of economic development, topography, and urban–rural structure, collectively forming the spatial distribution pattern of LES in the Chengdu Plain region.

**Author Contributions:** Methodology, L.Z. and J.Z.; Software, L.Z. and J.Z.; Formal analysis, L.Z.; Resources, W.P. and J.Z.; Writing—original draft, L.Z.; Writing—review & editing, W.P.; Supervision, W.P. All authors have read and agreed to the published version of the manuscript.

**Funding:** This research received no external funding.

**Data Availability Statement:** Data will be made available upon request. Landsat data were acquired from the USGS (https://earthexplorer.usgs.gov, accessed on 18 August 2022) and Environmental Sciences, Chinese Academy of Sciences (RESDC) (http://www.resdc.cn, accessed on 18 August 2022).

**Acknowledgments:** The authors thank the editors and anonymous referees for their valuable comments and suggestions, which helped improve the manuscript. The authors are pleased to thank NASA (https://search.earthdata.nasa.gov/search, accessed on 18 August 2022) and RESDC (https://www.resdc.cn/, accessed on 18 August 2022) for the provision of Landsat data. The funding

sources had no involvement in the collection, analysis, data interpretation, writing, or the decision to submit this study for publication.

**Conflicts of Interest:** The authors declare no competing interests.

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
