# Peer review of "Assessment of Land Ecological Security from 2000 to 2020 in the Chengdu Plain Region of China"

_land, doi:10.3390/land12071448_

Round 1

Reviewer 1 Report

Assessment of land ecological security from 2000 to 2020 in the Chengdu Plain region of China

Abstract

Line 12 – The sentence is too long and needs to be paraphrased

Line 17 - Pressure-State-Response (PSR) framework – is this the DPSIR framework? Or this is different

Authors should specify the datasets used in building this model

Line 23 - overall, the level of land ecological safety is relatively safe, but it exhibited a decreasing – what is level? The authors should present results that support these statements

Line 26 – this result should be reflected in the problem statement or opening statement

Introduction

Line 51 – 65 – this paragraph should be strengthened by providing more details on PSR and DPSIR models

Authors should unpack literature on land ecology and land ecological security  - more information should be provided

Line 125 – authors should include a statement that shows the importance of this study, given the gaps identified

Line 127 – the objectives need to be restated, there are not smart – they need to be restated. E.g. 1) construct and utilize?

Research area and methods

Line 142 – include citation for this statement and others that are hanging

Figure 1 – improve the map by including scale and coordinates, secondly what are the unit values of the DEM

Data sources

Line 174 – provide more details on data pre processing and quality of the datasets utilized in this study, but also limitations should be stated

Construction of ecological security evaluation system

Line 211 – Is this system or a model

Table 2 grading scale for soil stability assessment – the authors should clearly define between land use and cover – in normal cases – land cover does not include building sites

Line 256 - Landscape fragmentation index – the paragraphs should be rephrased. Also,  provide justification

Line 276 – table 3… what is unused land? Its important to define land use/cover types

Line 355 - Validation model for total ecosystem service value – this process should be strengthened to validate the model but as of now its weak and subjective

Line 391 – study on LES model – this seems not be a model but analytical parameters or tests

Line 403 – with the provided GDP values – please provide citations

Results and analysis

Line 413 – authors should provide similarities and differences in the indices results provided

Line 444 – area changes – this is not about trend but net changes. Authors should be rethink about this

Figure 4: Land ecological security – this legend labels need to be modified – what is unsafety?

Discussion

Authors should provide new findings that should be discussed. Secondly, the discussion should reflect the results but as per now, it is out of range

The paper has long sentences and many statements lack citations

Author Response

Manuscript revise letter

Thanks reviewers for your comments, which will help improve the quality of paper.

Reviewer #1: The paper has long sentences and many statements lack citations

1, Abstract: "The assessment of land ecological security (LES) aims to determine the impact of land use and human activities on the land ecosystem, with the ultimate goal of providing decision support and guidance for the protection and restoration of the health and functionality of the land ecosystem. "The sentence is too long and needs to be paraphrased

Revised: Thank reviewer for comments. The sentence has been revised, and the revised sentence has been incorporated into the paper and marked in red.

Abstract: The purpose of land ecological security (LES) assessment is to evaluate the influence of land use and human activities on the land ecosystem. Its ultimate objective is to offer decision-making assistance and direction for safeguarding and rejuvenating the well-being and effectiveness of the land ecosystem.

2, Abstract: "The model is built upon an index system developed using the Pressure-State-Response (PSR) framework ." Pressure-State-Response (PSR) framework – is this the DPSIR framework? Or this is different.Authors should specify the datasets used in building this model

Revised:Thank reviewer for comments. The PSR framework and the DPSIR framework are different. Below is my understanding of both frameworks and the reasons why I chose the PSR framework:

The PSR framework focuses on analyzing the pressures, state, and response measures of environmental issues. It starts with examining the pressures exerted by human activities on the environment, such as pollution, resource extraction, and land use. Then, it assesses the environmental state, including indicators like water quality, biodiversity, and land cover, to understand the impacts on the environment. Finally, based on the evaluation of pressures and state, corresponding measures and management strategies are proposed to alleviate pressures, improve the environmental state, or address environmental challenges.

On the other hand, the DPSIR framework is more comprehensive and covers four aspects: drivers, pressures, state, and responses. The DPSIR framework provides a more comprehensive analytical framework to better understand the entire process of environmental issues. In the DPSIR framework, drivers refer to the root causes of environmental problems, while pressures represent the manifestations of these problems. The state component assesses the environmental issues by evaluating changes in the environmental condition and the extent of the impact. Lastly, responses refer to the measures and strategies taken to address or mitigate the environmental problems.

Therefore, the main difference between the two frameworks is that the DPSIR framework is more comprehensive, considering various aspects of environmental issues, including root causes and the extent of the impact. The PSR framework, on the other hand, focuses more on the pressures, state, and response measures, emphasizing measures to alleviate pressures and improve the environmental state. However, in practical applications, due to the greater number of dimensions in the DPSIR framework, it introduces more subjectivity in judgment and requires more data. In regions like the Chengdu Plain with limited statistical data available for long-term analysis, the PSR framework simplifies the dimensionality issue, reduces subjective influences, and better reflects the relationship between environmental issues and human activities. Taking into consideration the strengths and weaknesses of both frameworks and the characteristics of the study area, the PSR framework is chosen as the assessment framework for land ecological security in the Chengdu Plain.

3, Abstract: "overall, the level of land ecological safety is relatively safe, but it exhibited a decreasing trend with 2010 as a turning point, and the lowest value was observed in that year" – what is level? The authors should present results that support these statements.

Revised:Thank reviewer for comments. Below is my explanation of the "level" and the data support for the statement:

Firstly, in this study, our evaluation model achieved an accuracy rate of 75.55% in precision validation. This indicates that our model can relatively accurately reflect the level of land ecological safety in the region. Utilizing this evaluation model, we assessed the spatial distribution of land ecological safety indices for five periods from 2000 to 2020. By correlating the indices with ecosystem service values, we classified land ecological safety into three levels: unsafe, relatively safe, and safe. After calculating the average index values and the proportion of each level's area for the five periods, we found that the average index values fell within the range of relatively safe, and the proportion of the relatively safe level's distribution area was the largest. Therefore, based on these data, this study concludes that the overall level of land ecological safety in the Chengdu Plain region is relatively high.

Furthermore, from a longitudinal perspective, this study compared the changes in the average index values and the proportion of each level's area for the five periods. Although the average index values consistently remained within the range of relatively safe, there were fluctuations within this range. The average index value decreased from 0.5385 in 2000 to 0.5177 in 2010, but starting from 2010, it began to rise and reached 0.5545 in 2020. Additionally, the area of unsafe regions increased from 2634.1 square kilometers in the period of 2000-2010 to 9874.02 square kilometers, but then decreased to 3915.24 square kilometers by 2020. Therefore, based on these findings, this study suggests that the land ecological safety status in the Chengdu Plain region exhibited a deteriorating trend from 2000 to 2010, followed by gradual improvement from 2010 to 2020.

The ecological security index is generally safe, with a mean value in the moderate safety range. It experienced a turning point in 2010, showing initial deterioration followed by improvement, mainly due to the transition between unsafe and relatively safe zones.

  • Abstract:"the distribution of the unsafe and safe zones exhibit opposite trends in terms of area changes, with the safe zones continuously increasing in size;"this result should be reflected in the problem statement or opening statement.

Revised:Thank reviewer for comments. After fully considering the opinions of the reviewers, I have removed the conclusion from the summary part and put it in the results and analysis of the following article as the statement of the issue of the change of land ecological security level.

5, Introduction:"The construction of an indicator system relies on frameworks such as PSR (Pressure-State-Response) and DPSIR (Driving forces-Pressure-State-Impact-Response), which guide the selection of indicators for evaluating a region. For instance, Cheng et al.(2022) developed an indicator system based on the PSR model and employed the fuzzy comprehensive evaluation method to generate the national ecological security status map for the year 2018[1]. The reliability of the model was validated using three surrogate indicators. Zhao et al.(2019) applied the PSR model to construct an indicator system and examined the evolving trend of ecological security patterns in Zhangye City[10]. Similarly, Shi et al.(2021) established an indicator system based on the DPSIR model and found that social and economic pressures were the primary factors contributing to the conversion of ecological security areas into unsafe areas in Bai Town of Erdaohe[11]. Furthermore, Cui et al.(2021) developed an indicator system using the DPSIR model and highlighted the importance of driving forces and responses in influencing ecological security in the urban agglomeration of the Yangtze River Delta[12]. These models have yielded significant research outcomes owing to their multidimensional and comprehensive nature."this paragraph should be strengthened by providing more details on PSR and DPSIR models.Authors should unpack literature on land ecology and land ecological security  - more information should be provided

Revised:Thank reviewer for comments. The paragraph has been revised, and the revised paragraph has been incorporated into the paper and marked in red.

The construction of an indicator system relies on frameworks such as PSR (Pressure-State-Response) and DPSIR (Driving forces-Pressure-State-Impact-Response), which guide the selection of indicators for evaluating a region. The PSR framework focuses on understanding the impact of human activities on ecosystems by examining pressures, states, and responses. It provides an analytical approach to environmental issues, emphasizing how human activities exert pressures on the environment, which in turn affect its state, and prompt societal responses to mitigate pressures and improve environmental conditions.For example, Cheng et al. (2022) developed an indicator system based on the PSR model and utilized the fuzzy comprehensive evaluation method to generate the national ecological security status map for the year 2018[1]. The reliability of the model was validated using three surrogate indicators. Similarly, Zhao et al. (2019) applied the PSR model to construct an indicator system and investigated the evolving trend of ecological security patterns in Zhangye City[10].On the other hand, the DPSIR framework considers a broader range of factors, including social, economic, and political drivers of environmental change, as well as how these drivers exert pressures, impact environmental states, and trigger societal and policy responses. In the context of ecological security. For instance, Shi et al.(2021) established an indicator system based on the DPSIR model and found that social and economic pressures were the primary factors contributing to the conversion of ecological security areas into unsafe areas in Bai Town of Erdaohe[11].Cui et al. (2021) developed an indicator system using the DPSIR model and emphasized the significance of driving forces and responses in influencing ecological security in the urban agglomeration of the Yangtze River Delta[12].These models have yielded significant research outcomes due to their multidimensional and comprehensive nature. By following the guidance of the PSR and DPSIR models, researchers can select relevant indicators based on different aspects of the models, enabling a comprehensive and systematic evaluation of the ecological security status in a given region.

Land ecology, as a subfield of ecology, places greater emphasis on the structure, function, and condition of land [13]. In the realm of land ecology, it is important to consider the human-land relationship and the associated ecological values that arise from it [14]. This integration of human-environment dynamics has led to the development of land ecological security assessment [15,16]. Contrasting with land ecology, land ecological security specifically focuses on the capacity of land use patterns to sustainably meet the diverse needs of human society [14].For instance, Cheng et al. (2022) conducted an assessment of China's land ecological security in 2018 by selecting 18 indicators that encompassed both human demands and land conditions [1]. In the context of rapid urbanization, Zhao et al. (2022) examined the impact of various scenarios of land development on land ecological security, taking into account the changing patterns of human demands [17] . Similarly, in their study on the land ecological security pattern in the Jinan region, Liu et al. (2022) underscored the evaluation of land ecological security status by integrating the ecological values provided by land with human demands [18].

[13]Tong, S., Zhiming, F., Yanzhao, Y., Yumei, L., & Yanjuan, W. (2018). Research on land resource carrying capacity: Progress and prospects. Journal of Resources and Ecology, 9(4), 331-340.

[14]Hua, Y.; Yan, M.; Limin, D. Land ecological security assessment for Bai autonomous prefecture of Dali based using PSR model--with data in 2009 as case. Energy Procedia 2011, 5, 2172–2177.

[15]Xu, L.; Yin, H.; Li, Z.; Li, S. Land ecological security evaluation of Guangzhou, China. Int. J. Environ. Res. Public Health 2014, 11, 10537–10558.

[16]Ye, X., Zou, C. X., Liu, G. H., Lin, N. F., & Xu, M. J. (2018). Main research contents and advances in the ecological security pattern. Acta Ecol. Sin, 38(10), 3382-3392.

[17]Liu, Jinhua,,Cao, Xiangyang,,Zhao, Lesong,,Dong, Guanglong & Jia, Kun.(2022).(Spatiotemporal Differentiation of Land Ecological Security and Its Influencing Factors: A Case Study in Jinan, Shandong Province, China).FRONTIERS IN ENVIRONMENTAL SCIENCE.FRONTIERS MEDIA SA,10.

[18]Liu, Jinhua,,Cao, Xiangyang,,Zhao, Lesong,,Dong, Guanglong & Jia, Kun.(2022).(Spatiotemporal Differentiation of Land Ecological Security and Its Influencing Factors: A Case Study in Jinan, Shandong Province, China).FRONTIERS IN ENVIRONMENTAL SCIENCE.FRONTIERS MEDIA SA,10.

6, Introduction:"It is evident that the Chengdu Plain region lacks a systematic LES evaluation system, which poses challenges in scientifically assessing the status and evolving trends of LES in the region[26-28]. Existing studies have often utilized counties as units[21,26] or 1-kilometer grid cells[23] for managing the ecological security of regional land resources, which fails to meet the need for more detailed management in the medium-scale research area of the Chengdu Plain region." 

1)authors should include a statement that shows the importance of this study, given the gaps identified.

2)the objectives need to be restated, there are not smart – they need to be restated. E.g. 1) construct and utilize?Research area and methods.

Revised:Thank reviewer for comments. The paragraph has been revised, and the revised paragraph has been incorporated into the paper and marked in red.

It is evident that the Chengdu Plain region lacks a systematic LES evaluation system, which poses challenges in scientifically assessing the status and evolving trends of LES in the region[26-28].Given this disparity, this study aims to address the gap by constructing a comprehensive set of LES indicators based on the PSR framework. These indicators will be used to evaluate the land ecological security status in the Chengdu Plain region, and their validity will be verified through the valuation of ecosystem services. The findings will contribute to a better understanding of the current state and evolving trends of LES in the region, providing a scientific basis for its sustainable development. Moreover, existing research predominantly focuses on ecological security management at the county level[21,26] or 1-kilometer grid cells[23], which fails to meet the demand for more detailed management in the medium-scale research area of the Chengdu Plain region. To address this issue, the research scale in this study has been narrowed down, and a grid cell size of 100 meters has been identified as an optimal balance between detailed management requirements and data redundancy.

7, Research area and methods:"The Chengdu Plain region, also known as the West Sichuan Plain, is located in the southwestern part of China and is the largest alluvial plain in the Southwest region. "include citation for this statement and others that are hanging.

Revised:Thank reviewer for comments. The paragraph has been revised, and the revised paragraph has been incorporated into the paper and marked in red.

The Chengdu Plain region, also known as the West Sichuan Plain, is located in the southwestern part of China and is the largest alluvial plain in the Southwest region[29]. Spanning between longitude 103°E and 105°E and latitude 29.5°N and 32°N, it covers an approximate area of 18,000 square kilometers. Geographically, the Chengdu Plain region appears flat internally, but it actually encompasses various landforms such as mountains, hills, and plains due to the sloping geological structure[29]. The region has a subtropical humid monsoon climate with distinct seasons and abundant precipitation, with an annual rainfall ranging from 1200 to 1600 millimeters[28].

The Chengdu Plain region is not only the economic center of Sichuan Province but also a densely populated area with a concentrated population and GDP[23]. In response to changes in domestic and international circumstances, promoting domestic circulation, serving the Yangtze River Economic Belt, and contributing to the Belt and Road Initiative, the concept of constructing the Chengdu-Chongqing Dual-city Economic Circle has been proposed, encompassing all cities in the Chengdu Plain region. This indicates that the Chengdu Plain region will continue to play a significant role as an economic center on a larger scale[30]. However, rapid economic development has resulted in significant human disturbances and impacts on the ecological environment. Therefore, it is crucial to assess the ecological security of land resources in the Chengdu Plain region to fulfill its responsibility as an ecological protection demonstration area in the upper reaches of the Yangtze River and as a central economic hub in the Midwest.

[23]Wang, J., Hou, L., He, X., Liu, T., & Deng, Y. (2022). Land use changes and their ecological and environmental effects in the Chengdu Plain Urban Agglomeration from 2000 to 2019. Bulletin of Soil and Water Conservation, (1), 360-368. doi:10.13961/j.cnki.stbctb.2022.01.047.

[28]Liu, Q., Xu, P., Wang, Y., Peng, P., & Liu, Y. (2018). Evaluation of ecosystem provisioning services value in Linpan of the Chengdu Plain. Ecological Economy, (5), 195-200.

[29]Sichuan Encyclopedia Editorial Board. (1997). Encyclopaedia of Sichuan. Sichuan Dictionary Press.

[30]Yang, J., Li, Y., & Wang, R. (2015). Sichuan-Chongqing Region: Strategic pivot of "One Belt, One Road" and the Yangtze River Economic Belt. Economic System Reform, (4), 58-64.

8, Figure 1 – improve the map by including scale and coordinates, secondly what are the unit values of the DEM

Revised:Thank reviewer for comments. The modified Fig as follow:

Figure 1. Location of the study area

9, Data sources: provide more details on data pre processing and quality of the datasets utilized in this study, but also limitations should be stated.Construction of ecological security evaluation system.

Revised:Thank reviewer for comments. The following is a specific introduction to using the data set and the built index system:

Data Details:

Table 1. Research Data (2000-2020).

Type

Data Name

Spatial resolution

Format

Source

Human Factors

Land use

30m

Raster

Chinese Academy of Sciences, Resource and Environmental Science Data Center

County-level administrative boundaries

\

Vector

Annual GDP density

1km

Raster

Raster

Annual population density

1km

Annual nighttime lights[31]

1km

Raster

https://dataverse.harvard.edu/dataset.xhtml?persistentld=doi:10.7910/DVN/GIYGJU

Annual fertilizer application amount

County-level

Panel data

Statistical Yearbook

Annual public financial expenditure

PM2.5

Washington University in St. Louis, Atmospheric Composition Analysis Group

Road

\

Vector

Open Street Map

Natural Factors

Annual average temperature

1km

Raster

National Earth System Science Data Center

Annual average precipitation

1km

Annual NPP

1km

Chinese Academy of Sciences, Resource and Environmental Science Data Center

Annual NDVI

1km

Soil texture

1km

Ecosystem service value

1km

DEM

30m

Geographic Spatial Data Cloud

(1)The land use data used in this study follows the "Classification Standard of Land Use" from the Third National Land Survey, ensuring consistency and compatibility with national land use classifications. Specifically, cultivated land primarily includes paddy fields and dryland; forest land mainly consists of forests, shrubs, and gardens; grassland includes alpine meadows and artificial lawns; water bodies encompass rivers, lakes, reservoirs, and other water-covered surfaces; construction land comprises urban and rural residential areas, industrial and mining areas, commercial and service areas, transportation areas, and other artificial surfaces; unused land refers to bare land surfaces not covered by vegetation or artificial structures.

(2)The county-level administrative boundary data used in this study were sourced from the "China Multi-Year County-Level Administrative Boundary Dataset" published by Xu Xinliang through the Resource and Environmental Science Data Registration and Publishing System. Specifically, the spatial distribution data of county-level administrative boundaries for the year 2018 were primarily selected for analysis.

(3)The spatial distribution of China's GDP at the kilometer grid level is derived from national county-level GDP statistical data. It takes into account the spatial interaction patterns between land use types, nighttime light intensity, population density data closely related to human activities, and GDP. The data is generated through spatial interpolation to create a 1km * 1km spatial grid dataset.

(4)In processing the spatial distribution of China's population at the kilometer grid level, the first step is to calculate the population distribution weights based on land use types, nighttime light intensity, and population density. These weights are then standardized considering the influences from the three aspects mentioned above. Next, the total weights of each county-level administrative unit are calculated. Finally, by combining the population counts on the unit weight grid with the distribution map of total weights, the spatial distribution of population is generated using raster spatial calculations.

(5)To assess the nighttime light intensity, annual nighttime light data is employed, which is derived from the Defense Meteorological Program-Operational Linescan System (DMSP-OLS) data. These data have been carefully corrected and improved by Wu Yizhen et al. and published in the journal Transactions on Geoscience and Remote Sensing. The dataset incorporates the "pseudoinvariant pixel" method to calibrate the DMSP-OLS data, considering the differences between the SUOMI National Polar-orbiting Partnership Visible Infrared Imaging Radiometer (SNPP-VIIRS) data and DMSP-OLS data.

(6)The data on annual fertilizer application and public fiscal expenditures were collected from the statistical yearbooks published by the Sichuan Provincial Statistics Bureau and the official websites of municipal governments. To achieve spatialization of panel data, in the case of fertilizer application data, this study averaged the fertilizer application amount within each county onto the arable land per square meter within the county. Compared to the approach taken by Wang Juan et al., who averaged it over the county area, the spatialization method used in this study for fertilizer application is more scientifically and logically justified. The panel data on public fiscal expenditures are compared based on the county area, reflecting the response intensity of county governments to land-related issues per square kilometer within their respective counties.

(7)The spatial distribution data of PM2.5 is sourced from the Atmospheric Composition Analysis Group at Washington University in St. Louis. We downloaded the data from their website, specifically opting for the highest precision data at a resolution of 0.01°*0.01°. The data was initially in .nc format and was subsequently converted to .tif format. We then aggregated and summarized the grid values to the respective administrative units for analysis.

(8)For analyzing the temperature patterns, the annual average temperature data is derived by averaging monthly temperature data to generate yearly data, providing a comprehensive understanding of the temperature conditions in the study area. By utilizing these datasets, this study aims to obtain accurate and reliable information for assessing the LES in the Chengdu Plain region.

(9)The spatial distribution dataset of annual vegetation index (NDVI) in China is based on continuous time series of SPOT/VEGETATION NDVI satellite remote sensing data. It is generated using the maximum value synthesis method to create an annual vegetation index dataset. The annual NPP (Net Primary Productivity) data is derived from the MOD17A3 dataset, with a temporal resolution of 1 year and a spatial resolution of 500 meters. It has been resampled to a 100-meter resolution grid for analysis.

(10)The soil texture data in China is obtained from the compilation of soil profile data based on the 1:1,000,000 soil type map and the second soil survey. The soil texture classification is based on the content of sand, silt, and clay particles. The data is categorized into three major classes: Sand, Silt, and Clay, and the content of different soil particles in each class is represented as a percentage.

(11)The spatial distribution dataset of ecosystem service values in China's terrestrial ecosystems is based on nationwide land ecosystem type remote sensing classification data. It refers to the ecosystem service value equivalence factor method proposed by Xie Gaodi et al. The dataset estimates the values of four major categories and eleven types of ecosystem services for the years 2000, 2005, 2010, 2015, and 2020.The four major categories of ecosystem services include provisioning services (food production, raw materials production, water supply), regulating services (gas regulation, climate regulation, environmental purification, hydrological regulation), supporting services (soil retention, nutrient cycling, biodiversity), and cultural services (aesthetic landscapes).The dataset provides spatially distributed information on the values of these ecosystem services across different years.

Data preprocessing:

First, all grid data were resampled to a spatial resolution of 100 meters. The row and column numbers of the grids were aligned with the 2020 land use grid data as the reference. The projection coordinate system was set to WGS_1984_UTM_47N for consistency.For the land use data, it was reclassified into six major categories: cropland, forestland, grassland, water bodies, built-up land, and unused land.The soil texture data was classified into six categories based on the international standard for soil texture classification within the study area.Panel data, such as PM2.5 and public fiscal expenditures, were linked to the administrative units and converted into grid data with the same resolution, row and column numbers, and projection coordinates.Subsequently, the data were processed according to the evaluation indicators set by the Pressure-State-Response (PSR) framework. The specific procedures are as follows:

(1) Soil stability

To quantify soil stability in the Chengdu Plain region, the soil stability assessment method proposed by Zhao et al.(2020) is employed in this study, utilizing ArcGIS software[36]. ArcGIS Pro, a powerful geospatial analysis tool, is used to reclassify the slope, aspect, and land use types in the Chengdu Plain region according to the categories specified in Table 2. After reclassification, a weighted summation approach is applied, considering the assigned weights for slope, aspect, and land use types. The weighted summation combines these factors to generate the final soil stability map for the Chengdu Plain region.

Table 2 Grading scale for soil stability assessment indicators

Weighting

Type

Levels

Score

0.3

Slope

<5°

10

5° - 8°

8

8° - 15°

7

15° - 25°

5

25° - 35°

3

>35°

1

0.3

Aspect

-1

5

0-90

10

90-270

1

270-360

10

0.4

Land use

Woodland

10

Grassland

8

Water area

6

Arable land

2

Building sites

4

(2) Landscape Diversity Index

Land use types are the fundamental units that constitute landscape ecosystems, and the stability of landscape structure and functionality is of great significance for the safety of land ecosystems [36]. The diversity and fragmentation of landscape structure constrain various ecological processes and impact the stability and biodiversity of regional ecosystems [36]. Landscape diversity refers to the diversity of different landscape types within a specific spatial extent. Drawing on studies by Nelson Katherine [37], Yu H. [38], and others, this study considers the use of Shannon's diversity index to characterize the landscape diversity in the Chengdu Plain region as relatively reasonable. The calculation formula for Shannon's diversity index is as follows:

Where SHDI is the Landscape Diversity Index, n represents the total number of landscape types in each grid, and Pi represents the proportion of area in each grid occupied by landscape type i .

[36]Lilin Zou, Jianying Wang, Mengdi Bai “Assessing spatial–temporal heterogeneity of China’s landscape fragmentation in 1980–2020”  Ecological Indicators 136 (2022) 108654               https://doi.org/10.1016/j.ecolind.2022.108654.

[37]Nelson, K. S., Patalee, B., & Yao, B. (2022). Higher landscape diversity associated with improved crop production resilience in Kansas, USA. Environmental Research Letters, 17(8), 84011.

[38]Yu, H., Liu, X., Ma, Q., Li, L., Wu, W., Qi, M., ... & Xu, Z. (2022). Nitrogen deposition drives response and recovery in the context of precipitation change and its reversal in an arid ecosystem. Journal of Geophysical Research: Biogeosciences, 127(9), e2022JG006828.

(3) Regional development index

According to Liu et al.(1996) proposal, a comprehensive land use index and a land use model expression are used to assign values to different land types (table 3), enabling the quantification of the degree of land use[39].

Table 3 Table of land use classification indices

Grading Index

1

2

3

4

Type of land use

Unused land

Woodland, water bodies, grassland

Arable land

Building sites

The regional development index is calculated using the following formula:

                              (2)

Where L is the composite index of regional development ;and Ai represents the percentage of area occupied by class i of the total area.

In addition to the human response to environmental problems, the environment itself has a certain capacity to regulate, with areas of higher net primary productivity having a greater capacity to regenerate areas of ecological damage; the more resilient the ecosystem, the greater the capacity to withstand and repair environmental damage.

(4) Ecosystem resilience

This paper uses the following formula to measure ecosystem resilience based on a reading of the relevant literature[40-43]:

                            (3)

Where ECORES is the ecosystem resilience, D is the landscape diversity index,Si is the percentage of the area of the i landscape type, and Pi is the resilience score of the i landscape type. The resilience score is mainly determined by primary productivity, vegetation cover or expert scoring. In this paper, the score is based on land cover type with reference to relevant studies in the Chengdu Plain region[44-46]. The results are shown in table 4.

Table 4 Ecological resilience of the Chengdu Plain region Pi score

Classification

Type of land use

Score

Description

1

Woodland

95

Woodlands play a decisive role in maintaining ecosystem resilience

2

Grassland

67

Grassland is maintained and managed to increase the resilience limits of the area

3

Water bodies

59

This category is of greater significance for maintaining the resilience of ecosystems and must be intensively managed and maintained

4

Arable land

37

This category is of some significance for maintaining ecosystem resilience and must be used with care

5

Building sites

21

This category is less significant for maintaining ecosystem resilience

6

Unused land

8

This category contributes very little to ecological resilience and is even very degraded

Note: The data in the table are corrected from the relevant results of Zuo(2022) .

Index system constructed based on the PSR framework:

Table 5 Ecological security evaluation indicator system

Target layer

Guideline layer

Indicator layer

Weighting

Pressure

Pollution emissions

Pesticide and fertilizer loads

0.0571

Road pollution

0.0143

Pollution in built-up areas

0.0143

PM2.5 concentration

0.0714

Human interference

Nighttime lighting data

0.0143

Population density

0.0429

GDP density

0.0286

Status

Climate

Average annual temperature

0.0429

Annual precipitation

0.0429

Soil

Soil stability

0.1

Soil texture

0.0571

Vegetation

NDVI

0.0857

Landscape

Landscape fragmentation index

0.0429

Response

Humanity

Per capita financial expenditure

0.0286

Regional Development Index

0.1286

Environment

NPP

0.1143

Ecosystem resilience

0.1143

10, Data sources:"To achieve this, a systematic approach is employed, encompassing the establishment of an index system, determination of index weights, construction of an evaluation model, model validation, and analysis of spatial patterns."Is this system or a model?

Revised:Thank reviewer for comments.In this study, we first selected appropriate indicators based on the Pressure-State-Response (PSR) framework to form an indicator system. Then, we combined the weights determined by the combination weighting method for each indicator, forming the land ecological security assessment model. Therefore, the indicators constitute the system, and the combination of indicator weights forms the model.

11, Table 2 grading scale for soil stability assessment – the authors should clearly define between land use and cover – in normal cases – land cover does not include building sites

Revised:Thank reviewer for comments.Below is my modification:

Land use refers to the manner and purpose of human utilization of land, involving the occupation and utilization of land by different types of human activities. It describes the division of land into different functional areas such as farmland, urban areas, industrial zones, forests, and grasslands, reflecting the management and utilization goals of human society towards land. Land use is closely related to human economic, social, and environmental activities and is an expression of human activities on land resources.

Land cover refers to the situation where the Earth's surface is covered by different types of natural elements (such as vegetation, water bodies, rocks, deserts) or artificial features (such as buildings, roads, farmland). It describes the physical surface characteristics and composition of different regions on the Earth's surface, reflecting the distribution and relative proportion of various elements on the Earth's surface. Land cover is typically based on spatial distribution and is used to understand the ecosystems and landform features of the Earth's surface.

The error here was indeed due to an input mistake, and the modified text is as follows::

Table 2. Grading scale for soil stability assessment indicators.

Weighting

Type

Levels

Score

0.3

Slope

<5°

10

5° - 8°

8

8° - 15°

7

15° - 25°

5

25° - 35°

3

>35°

1

0.3

Aspect

-1

5

0-90

10

90-270

1

270-360

10

0.4

Land use

Woodland

10

Grassland

8

Water area

6

Arable land

2

Building sites

4

12, Landscape fragmentation index – the paragraphs should be rephrased. Also,  provide justification.

Revised:Thank reviewer for comments.The paragraph has been revised, and the revised paragraph has been incorporated into the paper and marked in red.

(2) Landscape Diversity Index

Land use types are the fundamental units that constitute landscape ecosystems, and the stability of landscape structure and functionality is of great significance for the safety of land ecosystems [36]. The diversity and fragmentation of landscape structure constrain various ecological processes and impact the stability and biodiversity of regional ecosystems [36]. Landscape diversity refers to the diversity of different landscape types within a specific spatial extent. Drawing on studies by Nelson Katherine [37], Yu H. [38], and others, this study considers the use of Shannon's diversity index to characterize the landscape diversity in the Chengdu Plain region as relatively reasonable. The calculation formula for Shannon's diversity index is as follows:

Where SHDI is the Landscape Diversity Index, n represents the total number of landscape types in each grid, and Pi represents the proportion of area in each grid occupied by landscape type i .

[36]Lilin Zou, Jianying Wang, Mengdi Bai “Assessing spatial–temporal heterogeneity of China’s landscape fragmentation in 1980–2020”  Ecological Indicators 136 (2022) 108654               https://doi.org/10.1016/j.ecolind.2022.108654。

[37]Nelson, K. S., Patalee, B., & Yao, B. (2022). Higher landscape diversity associated with improved crop production resilience in Kansas, USA. Environmental Research Letters, 17(8), 84011.

[38]Yu, H., Liu, X., Ma, Q., Li, L., Wu, W., Qi, M., ... & Xu, Z. (2022). Nitrogen deposition drives response and recovery in the context of precipitation change and its reversal in an arid ecosystem. Journal of Geophysical Research: Biogeosciences, 127(9), e2022JG006828.

13, table 3… what is unused land? Its important to define land use/cover types

Revised:Thank reviewer for comments. Fallow land is a type of land use that refers to land that has not yet been covered by vegetation or developed for human use. It is characterized by a bare surface without vegetation or artificial structures, as observed in imagery.

14, Validation model for total ecosystem service value – this process should be strengthened to validate the model but as of now its weak and subjective.

Revised:Thank reviewer for comments. Now, I will show the whole verification process as follows:

First, theoretically determine the verification relationship between the total value of ecosystem services and ecological security:

Ecological services refer to the material products and intangible services that humans directly or indirectly obtain from the structure, processes, and functions of ecosystems, which are essential for sustaining life[47-48]. They mainly encompass four categories: provisioning services, regulating services, supporting services, and cultural services. Ecosystem services have long been employed by numerous scholars to investigate the identification of ecological security patterns. Scholars have successively demonstrated the correlation and inherent mechanisms between ecological security and ecosystem service values[49-51]. The representativeness of service values in reflecting the ecological security status has been confirmed by a considerable body of scholarly research [52-53]. The assessment of ecosystem service values has become an important basis for ecological environmental protection, ecological functional zoning, environmental economic accounting, and ecological compensation decision-making[54-55]. The total value of ecosystem services can to some extent reflect the health status of the structure and functions of an ecosystem. The higher the value of ecosystem services, the stronger the ecological carrying capacity, and the greater the ecological security. Furthermore, LES is defined by the relatively high value and stable quantity of ecosystem services provided by land resources. In this study, the total value of ecosystem services was selected as the validation model for assessing the ecological security of the Chengdu Plain region.

Zuo et al.(2002) proposed a confirmation relationship between the value of ecosystem services based on human needs and ecological security evaluation[60]. Later, researchers such as He et al.(2016), Li et al.(2020), Wang et al.(2022), Wang et al.(2019) adjusted the ecological security patterns based on the value of ecosystem services, confirming the scientific validation between ecosystem services and ecological security[61-65].

Subsequently, a feasibility validation was conducted. Initially, a highly recognized dataset of ecosystem service values was obtained. Using the Fishnet tool in ArcGIS Pro, a 100-meter interval grid was generated to sample the Land Ecological Security Index map and the Total Ecosystem Service Value map, resulting in a dataset of 242,392 records. Kendall Rank correlation analysis was performed using MATLAB, revealing a strong correlation coefficient of 0.4212, indicating a significant relationship between the two variables. However, while plotting the Land Ecological Security Index and the Ecosystem Service Value map, it was observed that their correlation patterns exhibited block-like distributions. Analysis of the Ecosystem Service Value dataset indicated that the values were concentrated within three threshold ranges, primarily around 0, 300, and 800. Moreover, certain areas exhibited extremely high values in the Total Ecosystem Service Value, making it challenging to achieve an ideal corresponding classification effect through dimensionless processing. In light of this, the study proceeded to classify the two datasets and calculate their correspondence through spatial overlay, considering the relationship between Ecosystem Service Value and Land Ecological Security levels. Specifically, areas with high Ecosystem Service Value generally exhibited higher ecological security, while areas with low Ecosystem Service Value tended to have weaker ecological security. By dividing the datasets into levels and performing spatial overlay calculations, a corresponding rate of 75.55% was achieved. However, due to the distinct three-segment distribution of the Ecosystem Service Value dataset, the critical values between segments were not clearly defined. To mitigate subjective segmentation, the study determined the classification breakpoints as positions where the correspondence rate was maximized between the two concentrated distribution areas. After exploring all possible breakpoints, the optimal breakpoint was identified, resulting in a correspondence rate of 75.55%. The final optimal breakpoint position is presented in the table below.

Table 6 Corresponding grading of ecological security and ecosystem service values

Ecological security level

Number

Standards

Ecosystem Services Classification

Number

Standards

Unsafe

1

≤ 0.45

Low value area

1

≤ 100

Safer

2

0.45-0.6

Median Zone

2

100-500

Safety

3

>0.6

High Value Area

3

> 500

15, study on LES model – this seems not be a model but analytical parameters or tests.

Revised:Thank reviewer for comments. It is widely recognized that the subjective interpretation of land ecological security definitions has led to individual variations in the selection of indicators among scholars, hindering the establishment of a widely accepted evaluation model despite the extensive research on land ecological security assessment. Therefore, this study attempts to propose such an evaluation model, but in order to enhance the persuasiveness of my model, it is necessary to validate its reliability using appropriate methods [18]. Model validation allows for an assessment of the correspondence between subjective perceptions of land ecological security and objective measures, thus determining the scientific and reliable nature of the land ecological security evaluation model.

Hence, to validate the reliability of my model, I conducted a spatial correspondence analysis between the evaluation results of the year 2020 and the corresponding ecosystem service value in the same year. A high spatial correspondence rate between the evaluation results and the ecosystem service value indicates the reliability and alignment of my evaluation model with the real-world situation. It is only with a reliable evaluation model that the subsequent analysis and interpretation of the land ecological security assessment results in the Chengdu Plain region can be conducted to gain a deeper understanding of its ecological security status.

In summary, the LES model serves as an evaluation model, and this section merely represents a validation of its reliability.

16, "Based on the GDP values, the top 25% of regions in terms of regional GDP were categorized as relatively developed areas, the next 25% were classified as relatively less developed areas, and the bottom 25% were designated as economically backward areas, while other townships were classified as economically less developed areas."with the provided GDP values – please provide citations.

Revised:Thank reviewer for comments. In this study, the GDP values used were derived from the normalized GDP density map, which was obtained by aggregating the values at the township level using the area tabulation tool. The GDP rankings were then determined based on this aggregated data. Due to space limitations, the data for GDP values will be provided as an attachment.

17, authors should provide similarities and differences in the indices results provided.

Revised:Thank reviewer for comments. Process of Creating Different Zoning Types:

GDP Zoning: The GDP density data was aggregated using the area tabulation tool to calculate the average values. The data was then sorted and divided into four categories based on a 25% interval.

Urban-Rural Zoning: The land use data was used to classify urban areas as urban construction land and categorize other land types as rural areas, thus creating an urban-rural division.

Topographic Zoning: The spatial distribution data of landforms at a scale of 1:1,000,000 in China was utilized. Plains and low-altitude plateaus were classified as plain areas, while high-altitude plateaus, farmland with elevation undulations, and low to moderate hills were classified as low hills and terrains. The remaining areas were designated as mountainous regions.

18, area changes – this is not about trend but net changes. Authors should be rethink about this.

Revised:Thank reviewer for comments. In response to the reviewer's feedback, we have revised the section accordingly. We have further highlighted the changing characteristics among different land ecological security zones, providing a comprehensive understanding of the dynamics within each zone. This division allows for a more nuanced analysis of the land ecological security in the Chengdu Plain, capturing both the overall trend and the specific variations among different zones. Additionally, we have ensured that the summary of the trend is adequately addressed, incorporating a comprehensive evaluation of the changes observed.

In terms of area changes, there is an inverse trend between the unsafe and safe areas in the Chengdu Plain region. The unsafe areas have been decreasing while the safe areas have consistently increased. From 2000 to 2010, the total area of unsafe land resources in the Chengdu Plain region was 2,634.1 km2, 3,515.65 km2, and 9,874.02 km2, respectively. The unsafe areas exhibited a rapid and continuous growth, with an annual increase of 724 km2. The period between 2005 and 2010 experienced the highest growth rate, reaching 1,271 km2 per year. By 2010, the area of unsafe land resources in the Chengdu Plain region exceeded one-fourth of the entire plain, indicating concerning ecological security conditions.

From 2010 to 2020, the area of unsafe areas decreased to 9,874.02 km2, 4,930.34 km2, and 3,915.24 km2, with an annual reduction rate of 596 km2. In the past five years, the reduction rate has slowed down to 203 km2 per year, but it has not yet reached the ecological security level of 2005. In contrast, the safe areas demonstrated an opposite trend, with respective areas of 25,730.97 km2, 24,899.98 km2, 17,506.23 km2, 20,922.94 km2, and 21,474.75 km2. After 2010, the safe areas initially decreased and then began to increase.

The area of ecological safe areas was 9,033.80 km2, 8,986.24 km2, 10,018.62 km2, 11,545.59 km2, and 12,008.81 km2. Overall, the trend of ecological safe areas showed a steady increase throughout the 20-year period, with a small decrease observed only between 2000 and 2005. The study was conducted at a refined scale using a 100-meter grid, which balanced the need for detailed management and minimized data redundancy.

19, Land ecological security – this legend labels need to be modified – what is unsafety?

Revised:Thank reviewer for comments.The modified Fig as follow:

Figure 4. Results of land ecological safety evaluation at the grid scale in the Chengdu Plain region.

Figure 5. Results of the evaluation of LES at the township scale in the Chengdu Plain region.

20, Authors should provide new findings that should be discussed. Secondly, the discussion should reflect the results but as per now, it is out of range.

Revised:Thank you for your valuable feedback. Based on your suggestions, I have incorporated a succinct summary of the research findings, highlighting the main discoveries and results. I have also provided potential explanations and hypotheses for different aspects of the research outcomes, discussing the practical implications of the results. Additionally, I have proposed potential avenues for improvement to address the limitations of this study. he modified discussion section is presented below:

4  Discussion

In response to the significant uncertainty in the current assessment of land ecological security, this paper proposes a comprehensive evaluation model that combines the advantages of subjective and objective weighting methods. By balancing the expert knowledge in the subjective weighting method and the data characteristics in the objective weighting method, the shortcomings of a single weighting method, such as excessive subjectivity and inconsistencies with reality, are avoided. Researchers such as Li Peiwu [53] recognized the limitations of the Analytic Hierarchy Process (AHP) and entropy weight method in the ecological security assessment of Shenzhen, and combined the two methods by assigning a preference coefficient of 0.5 to determine the combined weights, ultimately obtaining an ecological security assessment model for Shenzhen. Wei He [54] suggested that to avoid arbitrary subjective weights and contradictions between objective weights and practical experience, it is necessary to organically combine AHP with entropy weight method, ensuring a good balance between the two and achieving better alignment between indicator weights and reality. Yiran Wang [55] utilized principal component analysis and grey clustering method to classify and evaluate the forest ecological security levels of 11 provinces (municipalities) in the Yangtze River Economic Belt, and obtained favorable evaluation results. Xinchang Zhang [56] combined AHP with the improved grey relational TOPSIS method to accurately evaluate the ecological security status of land in mining cities. The studies conducted by these scholars have confirmed that the combination of subjective and objective weighting methods can yield evaluation results that are more consistent with the actual situation. Therefore, based on the conventional AHP method, this paper integrates the knowledge and experience of multiple experts to obtain subjective weights, and combines them with the entropy weight method that reflects data characteristics, ultimately achieving a unified set of indicator weights.

To validate the level of Land Ecological Security (LES), this paper introduces, for the first time, a verification model that utilizes the widely recognized and fundamental measure of Ecological System Service (ESS) value, which is commonly used as a basis for ecological conservation. The relationship between the ESS value based on human demands and the evaluation of ecological security was initially proposed by Zuo Wei [57]. Subsequently, researchers such as He Lin [58], Li Can [59], Wang Zhengwei [60], and Wang Yun [61] adjusted the ecological security pattern based on ESS value, confirming the scientific correlation between ESS value and ecological security. In comparison to the approach taken by Cheng H et al. [1], which validates the model through individual corresponding dimensions, using ESS value as the ultimate validation model for assessing land resource ecological security provides a more intuitive reflection of the overall reliability of the model. Furthermore, compared to the methods employed by Zhao Liting [19], Liu Chenli [21], and others, who use multidimensional indicators for model verification, adopting a comprehensive single indicator allows for a more comprehensive consideration of the scope and provides clearer and more intuitive verification results.

Furthermore, the spatial clustering of the evaluation scores for the pressure, state, and response dimensions was effectively demonstrated using global Moran's I and local Moran's I. The spatial differentiation of economic development level, topography, and urban-rural structure was analyzed using the Mann-Whitney U test and Kruskal-Wallis H test. Due to the influence of topography, cities in the Chengdu Plain area are mainly distributed in the plain regions, resulting in the concentration of industries and population in these areas. However, when the concentration of population and industries becomes excessive and construction land encroaches upon ecological land, the regional ecological pressure becomes too high, leading to ecological insecurity. In the low hilly regions, cities are unable to concentrate in a contiguous manner due to topographical constraints, resulting in an intermixing of urban and rural areas. In this case, the distribution of population and industries is appropriately balanced, leading to a certain equilibrium between ecological pressure and carrying capacity, resulting in a relatively secure ecological state. In mountainous areas, the significant terrain variations make the region unsuitable for human production activities. Therefore, although the ecological state in these areas is generally average, the low human-induced pressure contributes to an ecological security status.

The study also found that the Chengdu Plain region as a whole is in a relatively secure ecological state. However, within this relatively secure state, there has been a turning point around 2010, with a trend of initial deterioration followed by improvement in the land ecological security. The main cause of this deterioration and subsequent improvement is the change in the quantity of areas categorized as relatively secure and ecologically insecure. During the period from 2000 to 2010, extensive conversion of farmland, grassland, and woodland into construction land occurred due to the needs of economic development. Additionally, the presence of unused land resulting from improper construction planning severely undermined regional land ecological security. Consequently, a significant number of areas classified as relatively secure experienced a shift to ecologically insecure areas, particularly during the period of rapid economic development from 2005 to 2010 in the Chengdu Plain. These changes were primarily concentrated around urban land in various districts and counties, as confirmed by the Mann-Whitney U test and Kruskal-Wallis H test. From 2010 to 2020, there was an increased emphasis on ecological environment protection in national policies, leading to greater attention from various sectors. Furthermore, the construction of ecological conservation zones in the upper reaches of the Yangtze River, along with the shift from extensive to intensive economic development in the Chengdu Plain region, contributed to a slowdown in urban land expansion, the implementation of land restoration projects (such as returning farmland to forest and grass), and the development of eco-friendly cities. As a result, the area of ecological land in the Chengdu Plain gradually increased. The progress of ecological restoration projects led to the transformation of some ecologically insecure areas into relatively secure areas and even into ecologically safe areas.

In summary, this study provides a comprehensive model for assessing land ecological security, successfully evaluating the land ecological security level in the Chengdu Plain region. In terms of balancing economic development and ecological protection, there is an interactive relationship between ecosystem services and ecological security levels. Economic development relies on the resources and services provided by ecosystems, but excessive economic activities can potentially cause damage and pressure on ecosystems. Therefore, this study makes the relationship between ecosystem services and land ecological security explicit, enabling a better understanding of the vital contribution of ecosystems to the economy. It suggests incorporating this value into economic decision-making to achieve coordinated development between economic growth and ecological protection. The findings of this research have important implications for decision-makers in formulating land ecological protection policies and management decisions.

However, research need to be further improved and refined.Although this study constructs an indicator system based on the PSR model, taking into account various dimensions and aspects, the subjective nature of indicator selection introduces individual characteristics. For instance, GDP values have been used as indicators in the pressure dimension[65], as well as in the response dimension by considering industrial segmentation[66-67], and even as indicators in the state dimension by normalizing them per capita[65,68]. NDVI values, which reflect regional vegetation coverage, are commonly used for evaluating the state dimension[69], and some have further calculated forest coverage or green space coverage as indicators in the response dimension[65]. Furthermore, the completeness of regional data can also affect indicator selection. In the process of selecting data for the pressure and response dimensions, the statistical criteria and indicators vary across different cities and years. Consequently, energy consumption data, pollution control investment data, and education levels, among others, were unavailable, and alternative indicators had to be chosen to indirectly reflect such data. Thus, the 17 selected indicators may not comprehensively represent the ecological security status of Chengdu Plain region, highlighting the need for further research in indicator system selection and the incorporation of logical and methodological approaches from operations research and management science to enhance objectivity.

Inconsistent spatial resolution impacts the final validation accuracy. When evaluating the ecological security of land resources in Chengdu Plain region, this study considered the small-scale nature of the region and found that using 1-kilometer grid data may not meet the requirements for fine-scale management of land resources. Therefore, multiple resolutions were explored, and ultimately, 100-meter grid data were chosen for the study. However, the validation data used in this study were obtained from the Data Center for Resources and Environmental Sciences of the Chinese Academy of Sciences, and the ecosystem service value data provided on their website had a resolution of 1 kilometer. Although bilinear interpolation was employed to downscale the data, the precision still falls short of the validation data's accuracy. As a result, in regions with small aggregation ranges of ecological security evaluation results or transitional zones between two evaluation results, the downscaled ecosystem service value data fails to accurately represent the data details, leading to misclassifications in certain data details.

Reviewer 2 Report

[Land] Manuscript ID: land-2495599

Thank you very much for giving me the opportunity to review this manuscript.

Ecological land conservation had become more and more important with the rapid development of urban areas together with the human-induced issues on natural lands. This paper assessed the land ecological security in the Chengdu Plain region of China between 2000 and 2020. The paper proposed an evaluation model to evaluate the land ecological safety based on the previous Pressure-State-Response (PSR) framework. The paper also used statistical methods to understand the distribution patterns of regional land ecological safety levels. This was an innovative approach, and the results were good.

Each section is balanced and clearly written. The introduction gives enough background information on the concept of ecological security and the methods for its evaluation. The technical, structural, and formal quality of the manuscript is good. The methods used in this paper are a useful contribution to the literature on our understanding of the patterns of ecological land safety.

The paper is well-structured, written, and referenced. The figures and tables are clear and understandable. But some tables are split into two pages, which is hard to follow (e.g., Page 8, Line 292) as well as some of the captions of tables are left on the previous page (e.g., Page 13, Line 470-471). I would suggest to tied up these tables to present them to your audience in a better and clear way. I would also suggest authors plot the figure for the location of the study area, for the ease of international audience.

In the discussion section, I would add more discussion of the strengths and weaknesses of the methodology. Finally, in this section, I believe that it is necessary to discuss the relationships between the ecosystem service's total value and the level of ecological security in balancing economic development and ecological conservation.

Author Response

Manuscript revise letter

Thanks reviewers for your comments, which will help improve the quality of paper.

Reviewer #1: I would suggest to tied up these tables to present them to your audience in a better and clear way. I would also suggest authors plot the figure for the location of the study area, for the ease of international audience.

Revised:Thank you very much for pointing out the problem, which is indeed something I did not consider in my previous indicator mapping. Here is my revised diagram, and I will modify the table in the original text.

Figure 1. Location of the study area.

2, In the discussion section, I would add more discussion of the strengths and weaknesses of the methodology. Finally, in this section, I believe that it is necessary to discuss the relationships between the ecosystem service's total value and the level of ecological security in balancing economic development and ecological conservation.

Revised:Thank you for your valuable feedback. According to your suggestion, I have added the discussion on the advantages and disadvantages of methods in the discussion section and the discussion on the relationship between ecosystem service value and land ecological security, and the balance between economic construction and ecological protection. The following is my revised discussion. I marked the two points mentioned above in red.

4  Discussion

In response to the significant uncertainty in the current assessment of land ecological security, this paper proposes a comprehensive evaluation model that combines the advantages of subjective and objective weighting methods. By balancing the expert knowledge in the subjective weighting method and the data characteristics in the objective weighting method, the shortcomings of a single weighting method, such as excessive subjectivity and inconsistencies with reality, are avoided. Researchers such as Li Peiwu [53] recognized the limitations of the Analytic Hierarchy Process (AHP) and entropy weight method in the ecological security assessment of Shenzhen, and combined the two methods by assigning a preference coefficient of 0.5 to determine the combined weights, ultimately obtaining an ecological security assessment model for Shenzhen. Wei He [54] suggested that to avoid arbitrary subjective weights and contradictions between objective weights and practical experience, it is necessary to organically combine AHP with entropy weight method, ensuring a good balance between the two and achieving better alignment between indicator weights and reality. Yiran Wang [55] utilized principal component analysis and grey clustering method to classify and evaluate the forest ecological security levels of 11 provinces (municipalities) in the Yangtze River Economic Belt, and obtained favorable evaluation results. Xinchang Zhang [56] combined AHP with the improved grey relational TOPSIS method to accurately evaluate the ecological security status of land in mining cities. The studies conducted by these scholars have confirmed that the combination of subjective and objective weighting methods can yield evaluation results that are more consistent with the actual situation. Therefore, based on the conventional AHP method, this paper integrates the knowledge and experience of multiple experts to obtain subjective weights, and combines them with the entropy weight method that reflects data characteristics, ultimately achieving a unified set of indicator weights.In the combined subjective-objective method of composite weighting, in order to avoid significant deviations in the weights of individual indicators caused by excessive entropy values of certain data in the objective weighting method, data cleaning is required. Additionally, to prevent data redundancy resulting from a large volume of data, data lightweighting is necessary. The complexity of data processing is further heightened by multiple steps such as normalization and standardization of indicators. Furthermore, the determination of trade-offs between weights obtained from subjective and objective sources requires more reliable data and accurate expert opinions, as well as careful handling of different expert perspectives and weight considerations.

To validate the level of Land Ecological Security (LES), this paper introduces, for the first time, a verification model that utilizes the widely recognized and fundamental measure of Ecological System Service (ESS) value, which is commonly used as a basis for ecological conservation. The relationship between the ESS value based on human demands and the evaluation of ecological security was initially proposed by Zuo Wei [57]. Subsequently, researchers such as He Lin [58], Li Can [59], Wang Zhengwei [60], and Wang Yun [61] adjusted the ecological security pattern based on ESS value, confirming the scientific correlation between ESS value and ecological security. In comparison to the approach taken by Cheng H et al. [1], which validates the model through individual corresponding dimensions, using ESS value as the ultimate validation model for assessing land resource ecological security provides a more intuitive reflection of the overall reliability of the model. Furthermore, compared to the methods employed by Zhao Liting [19], Liu Chenli [21], and others, who use multidimensional indicators for model verification, adopting a comprehensive single indicator allows for a more comprehensive consideration of the scope and provides clearer and more intuitive verification results. Although there is a corresponding relationship between the value of ecosystem services and land ecological security, it is important to note that this relationship is not strictly linear. This is fully reflected in the maximum validation accuracy of up to 75.55%. While some of the errors in between can be attributed to inconsistencies in the original data resolution, there are also differences in the definitions of the two concepts. For example, in the case of unused land, the ecosystem service value may be calculated as zero, but it does not necessarily indicate an unsafe condition in terms of land ecological security. This is also related to human demands imposed on that land. Fragile land ecosystems, which receive high attention and protection and have lower human demands, may exhibit convergent changes in land ecological security influenced by the surrounding environment. This is particularly evident in the unused land on the western side of Longmen Mountain.

Thinking on the relationship between ecosystem service value and land ecological security reveals the interactive balance between economic development and ecological protection. The potential conflict between ecological security and economic development stems from the competitive demand for natural resources. Economic activity often puts pressure on ecosystems, leading to degradation and a decline in ecological security. However, ecological security is also crucial for maintaining long-term economic development. Environmental degradation can have adverse effects on various economic sectors such as agriculture, tourism and public health. Thus, achieving a balance between the two requires dealing with potential conflicts and exploring synergies, which involve the implementation of policies and regulations that promote sustainable resource management, conservation measures, and incorporate ecological considerations into economic decision-making processes. By adopting an ecosystem-based approach, policymakers can identify win-win solutions while strengthening ecological security and supporting economic growth.

Furthermore, the spatial clustering of the evaluation scores for the pressure, state, and response dimensions was effectively demonstrated using global Moran's I and local Moran's I. The spatial differentiation of economic development level, topography, and urban-rural structure was analyzed using the Mann-Whitney U test and Kruskal-Wallis H test. Due to the influence of topography, cities in the Chengdu Plain area are mainly distributed in the plain regions, resulting in the concentration of industries and population in these areas. However, when the concentration of population and industries becomes excessive and construction land encroaches upon ecological land, the regional ecological pressure becomes too high, leading to ecological insecurity. In the low hilly regions, cities are unable to concentrate in a contiguous manner due to topographical constraints, resulting in an intermixing of urban and rural areas. In this case, the distribution of population and industries is appropriately balanced, leading to a certain equilibrium between ecological pressure and carrying capacity, resulting in a relatively secure ecological state. In mountainous areas, the significant terrain variations make the region unsuitable for human production activities. Therefore, although the ecological state in these areas is generally average, the low human-induced pressure contributes to an ecological security status.

The study also found that the Chengdu Plain region as a whole is in a relatively secure ecological state. However, within this relatively secure state, there has been a turning point around 2010, with a trend of initial deterioration followed by improvement in the land ecological security. The main cause of this deterioration and subsequent improvement is the change in the quantity of areas categorized as relatively secure and ecologically insecure. During the period from 2000 to 2010, extensive conversion of farmland, grassland, and woodland into construction land occurred due to the needs of economic development. Additionally, the presence of unused land resulting from improper construction planning severely undermined regional land ecological security. Consequently, a significant number of areas classified as relatively secure experienced a shift to ecologically insecure areas, particularly during the period of rapid economic development from 2005 to 2010 in the Chengdu Plain. These changes were primarily concentrated around urban land in various districts and counties, as confirmed by the Mann-Whitney U test and Kruskal-Wallis H test. From 2010 to 2020, there was an increased emphasis on ecological environment protection in national policies, leading to greater attention from various sectors. Furthermore, the construction of ecological conservation zones in the upper reaches of the Yangtze River, along with the shift from extensive to intensive economic development in the Chengdu Plain region, contributed to a slowdown in urban land expansion, the implementation of land restoration projects (such as returning farmland to forest and grass), and the development of eco-friendly cities. As a result, the area of ecological land in the Chengdu Plain gradually increased. The progress of ecological restoration projects led to the transformation of some ecologically insecure areas into relatively secure areas and even into ecologically safe areas.

In summary, this study provides a comprehensive model for assessing land ecological security, successfully evaluating the land ecological security level in the Chengdu Plain region. However, research need to be further improved and refined.Although this study constructs an indicator system based on the PSR model, taking into account various dimensions and aspects, the subjective nature of indicator selection introduces individual characteristics. For instance, GDP values have been used as indicators in the pressure dimension[65], as well as in the response dimension by considering industrial segmentation[66-67], and even as indicators in the state dimension by normalizing them per capita[65,68]. NDVI values, which reflect regional vegetation coverage, are commonly used for evaluating the state dimension[69], and some have further calculated forest coverage or green space coverage as indicators in the response dimension[65]. Furthermore, the completeness of regional data can also affect indicator selection. In the process of selecting data for the pressure and response dimensions, the statistical criteria and indicators vary across different cities and years. Consequently, energy consumption data, pollution control investment data, and education levels, among others, were unavailable, and alternative indicators had to be chosen to indirectly reflect such data. Thus, the 17 selected indicators may not comprehensively represent the ecological security status of Chengdu Plain region, highlighting the need for further research in indicator system selection and the incorporation of logical and methodological approaches from operations research and management science to enhance objectivity.

Inconsistent spatial resolution impacts the final validation accuracy. When evaluating the ecological security of land resources in Chengdu Plain region, this study considered the small-scale nature of the region and found that using 1-kilometer grid data may not meet the requirements for fine-scale management of land resources. Therefore, multiple resolutions were explored, and ultimately, 100-meter grid data were chosen for the study. However, the validation data used in this study were obtained from the Data Center for Resources and Environmental Sciences of the Chinese Academy of Sciences, and the ecosystem service value data provided on their website had a resolution of 1 kilometer. Although bilinear interpolation was employed to downscale the data, the precision still falls short of the validation data's accuracy. As a result, in regions with small aggregation ranges of ecological security evaluation results or transitional zones between two evaluation results, the downscaled ecosystem service value data fails to accurately represent the data details, leading to misclassifications in certain data details.
